# APP interacts with LRP4 and agrin to coordinate the development of the neuromuscular junction in mice

Hong Y Choi[1†], Yun Liu[2†], Christian Tennert[1‡], Yoshie Sugiura[2], Andromachi Karakatsani[3], Stephan Kröger[3], Eric B Johnson[1], Robert E Hammer[4], Weichun Lin[2*], Joachim Herz[1,2,5,6*]

[1]Department of Molecular Genetics, University of Texas Southwestern Medical Center, Dallas, United States; [2]Department of Neuroscience, University of Texas Southwestern Medical Center, Dallas, United States; [3]Department of Physiological Genomics, Ludwig-Maximilians-Universität München, München, Germany; [4]Department of Biochemistry, University of Texas Southwestern Medical Center, Dallas, United States; [5]Department of Neurology and Neurotherapeutics, University of Texas Southwestern Medical Center, Dallas, United States; [6]Center for Neuroscience, Albert-Ludwigs-Universität Freiburg, Freiburg, Germany

*For correspondence: weichun. lin@utsouthwestern.edu (WL); joachim.herz@utsouthwestern. edu (JH)

†These authors contributed equally to this work

‡Present address: Department of Operative Dentistry and Periodontology, University School and Dental Hospital, Albert-Ludwigs-Universität Freiburg, Freiburg, Germany

Competing interests: The authors declare that no competing interests exist.

**Abstract** ApoE, ApoE receptors and APP cooperate in the pathogenesis of Alzheimer's disease. Intriguingly, the ApoE receptor LRP4 and APP are also required for normal formation and function of the neuromuscular junction (NMJ). In this study, we show that APP interacts with LRP4, an obligate co-receptor for muscle-specific tyrosine kinase (MuSK). Agrin, a ligand for LRP4, also binds to APP and co-operatively enhances the interaction of APP with LRP4. In cultured myotubes, APP synergistically increases agrin-induced acetylcholine receptor (AChR) clustering. Deletion of the transmembrane domain of LRP4 (LRP4 ECD) results in growth retardation of the NMJ, and these defects are markedly enhanced in APP[−/−];LRP4[ECD/ECD] mice. Double mutant NMJs are significantly reduced in size and number, resulting in perinatal lethality. Our findings reveal novel roles for APP in regulating neuromuscular synapse formation through hetero-oligomeric interaction with LRP4 and agrin and thereby provide new insights into the molecular mechanisms that govern NMJ formation and maintenance.

## Introduction

Apolipoprotein E (ApoE) ε4 genotype is the strongest and most prevalent risk factor for late-onset Alzheimer's disease (AD) (*Corder et al., 1993*; *Schmechel et al., 1993*). ApoE binds to and modulates the function of ApoE receptors, a family of LDL receptor-related proteins (LRPs) and postsynaptic signal transducers that regulate glutamatergic neurotransmission in the CNS (*Weeber et al., 2002*; *Beffert et al., 2005*; *Herz and Chen, 2006*) and the formation of the neuromuscular synapse in the periphery (*Weatherbee et al., 2006*; *Zhang et al., 2008*; *Kim et al., 2008b*).

Mutations in the amyloid precursor protein (APP) cause early-onset AD (*Goate et al., 1991*). APP and its pathogenic cleavage product, β-amyloid, physically and functionally interact with ApoE receptors on multiple levels (*Kounnas et al., 1995*), by regulating the trafficking and processing of APP (*Ulery et al., 2000*; *Pietrzik et al., 2002*; *Andersen et al., 2005*; *Hoe et al., 2005*; *Pietrzik and Jaeger, 2008*; *Marzolo and Bu, 2009*), mediating amyloid clearance (*Andersen et al., 2005*; *Deane et al., 2008*) and by preventing amyloid-induced synaptic suppression at the synapse (*Durakoglugil et al., 2009*). Intriguingly, APP is also expressed at the neuromuscular junction (*Akaaboune et al., 2000*) and

**eLife digest** One of the hallmarks of Alzheimer's disease is the formation of plaques in the brain by a protein called β-amyloid. This protein is generated by the cleavage of a precursor protein, and mutations in the gene that encodes amyloid precursor protein greatly increase the risk of developing a familial, early-onset form of Alzheimer's disease in middle age. Individuals with a particular variant of a lipoprotein called ApoE (ApoE4) are also more likely to develop Alzheimer's disease at a younger age than the rest of the population. Due to its prevalence—approximately 20% of the world's population are carriers of at least one allele—ApoE4 is the single-most important risk factor for the late-onset form of Alzheimer's disease.

Amyloid precursor protein and the receptors for ApoE—in particular one called LRP4—are also essential for the development of the specialized synapse that forms between motor neurons and muscles. However, the mechanisms by which they, individually or together, contribute to the formation of these neuromuscular junctions are incompletely understood.

Now, Choi et al. have shown that amyloid precursor protein and LRP4 interact at the developing neuromuscular junction. A protein called agrin, which is produced by motor neurons and which must bind to LRP4 to induce neuromuscular junction formation, also binds directly to amyloid precursor protein. This latter interaction leads to the formation of a complex between LRP4 and amyloid precursor protein that robustly promotes the formation of the neuromuscular junction. Mutations that remove the part of LRP4 that anchors it to the cell membrane weaken this complex and thus reduce the development of neuromuscular junctions in mice, especially if the animals also lack amyloid precursor protein.

These three proteins thus seem to influence the development and maintenance of neuromuscular junctions by regulating the activity of a fourth protein, called MuSK, which is present on the surface of muscle cells. Activation of MuSK by agrin bound to LRP4 promotes the clustering of acetylcholine receptors in the membrane, which is a crucial step in the formation of the neuromuscular junction. Intriguingly, Choi et al. have now shown that amyloid precursor protein can, by interacting directly with LRP4, also activate MuSK even in the absence of agrin, albeit only to a small extent.

The work of Choi et al. suggests that the complex formed between agrin, amyloid precursor protein and LRP4 helps to focus the activation of MuSK, and thus the clustering of acetylcholine receptors, to the site of the developing neuromuscular junction. Since all four proteins are also found in the central nervous system, similar processes might well be at work during the development and maintenance of synapses in the brain. Further studies of these interactions, both at the neuromuscular junction and in the brain, should shed new light on both normal synapse formation and the synaptic dysfunction that is seen in Alzheimer's disease.

APP family members are required for normal formation and function of the neuromuscular junction (*Torroja et al., 1999*; *Wang et al., 2005*, *2009*), although the underlying mechanisms remain unclear. APP family members have also been shown to contribute to synaptic function in the CNS (*Weyer et al., 2011*).

Although APP occupies such a central role in the pathogenesis of AD, its physiological functions and how they relate to the disease process on the molecular level remains poorly understood. ApoE ε4 genotype strongly predisposes to an earlier age of AD onset, increasing the relative risk in individuals >65 years of age by approximately 10-fold (*Corder et al., 1993*; *Schmechel et al., 1993*). APP and ApoE interact with LRPs, thus suggesting a role of ApoE receptors in AD pathogenesis (*Herz and Beffert, 2000*). Besides serving as cargo receptors that can mediate the endocytosis of ApoE containing lipoprotein particles, ApoE receptors have also been shown to function as signal transducers that regulate essential signaling pathways during development and in the adult organism, where they control glutamate receptor function, synaptic plasticity, memory and learning (reviewed in *Herz and Chen, 2006*). Examples include brain development (LRP8, VLDLR), vascular development and maintenance (LRP1), kidney and neuromuscular junction formation (LRP4), and others (reviewed in *Dieckmann et al., 2010*).

The NMJ is a specialized peripheral cholinergic synapse (*Hall and Sanes, 1993*; *Sanes and Lichtman, 2001*; *Wu et al., 2010*). Impaired cholinergic neurotransmission has been implicated in AD and this forms the conceptual basis for the therapeutic use of cholinesterase inhibitors. Both APP and

LRP4 are required for normal NMJ formation, and the LRP4 ligand agrin is abundantly expressed in the CNS (*Burgess et al., 2000*; *O'Connor et al., 1994*; *Stone and Nikolics, 1995*). Furthermore, MuSK localizes to excitatory synapses (*Ksiazek et al., 2007*) and has been shown to mediate cholinergic responses, synaptic plasticity and memory formation (*Garcia-Osta et al., 2006*), while agrin was found to regulate synaptogenesis (*McCroskery et al., 2009*) and prevent synapse loss in the cortex (*Ksiazek et al., 2007*). These intriguing similarities between MuSK, agrin and ApoE receptors in synaptic maintenance and function prompted us to explore the NMJ as a model system on which potentially novel functional interactions between APP and ApoE receptors could be investigated. We found that LRP4 and APP physically bind to each other and that APP can also bind agrin. Ligation of LRP4 by immobilized APP can activate MuSK in the absence of neural agrin. Simultaneous and stoichiometric interaction of the three proteins favors the formation of a hetero-oligomeric complex, which may serve to focus the formation of the NMJ at the surface of the myotube.

## Results

### NMJ formation in mice lacking membrane-anchored LRP4

A common mechanism by which ApoE receptors directly transduce or indirectly modulate extracellular signals involves the interaction of adaptor proteins with their intracellular domain (ICD) (*Trommsdorff et al., 1998*, *1999*; *Gotthardt et al., 2000*). We therefore generated three mutant *Lrp4* alleles in mice to test which domains of LRP4 are required for NMJ development. Consistent with the previous report that mice with point mutations in *Lrp4* die at birth (*Weatherbee et al., 2006*), mice carrying a novel *Lrp4* null allele, which we generated by deleting exon 1 of murine *Lrp4* (*Figure 1A*), also die perinatally from a complete failure to form NMJs (*Figure 2A*). By contrast, mice carrying an *Lrp4* allele encoding a truncated receptor consisting of only the extracellular domain (ECD), but lacking the transmembrane segment and intracellular domain (ICD), are viable (*Johnson et al., 2005*), indicating that at least partially functional NMJs must form and that thus neither membrane anchoring of the LRP4 ECD nor its ICD is absolutely required for the formation of the NMJ (*Dietrich et al., 2010*; *Gomez and Burden, 2011*). To further test this hypothesis, we first set out to determine to what extent membrane anchoring of the LRP4 receptor is required for NMJ formation. We examined NMJs of *Lrp4*$^{ECD/ECD}$ mice, in which we introduced a stop codon immediately before the transmembrane segment. This results in the expression of LRP4 with the extracellular domain, but without the transmembrane domain or the intracellular domain (*Figure 1B*) (*Johnson et al., 2005*).

One of the hallmarks of NMJ formation is that presynaptic nerve terminals progressively accumulate synaptic vesicles in juxtaposition with AChR clusters that accumulate on the postsynaptic membrane (*Sanes and Lichtman, 1999*; *Wu et al., 2010*). We therefore double-labeled embryonic diaphragm muscles (E16.5) with anti-syntaxin antibodies and α-bungarotoxin to detect presynaptic nerve terminals and postsynaptic AChRs, respectively. In contrast to the complete absence of pre- and postsynaptic differentiation in *Lrp4* null mice (*Figure 2A*, middle row), neuromuscular synapses were present in the E16.5 *Lrp4*$^{ECD/ECD}$ embryos. Presynaptic terminals were properly juxtaposed to postsynaptic AChR clusters (*Figure 2A*, bottom row), which aligned along the central region of the muscle, although the density of the AChR appeared reduced. These results indicate an initial formation of the NMJs in E16.5 *Lrp4*$^{ECD/ECD}$ embryos.

### Membrane anchoring, but not the intracellular domain (ICD) of LRP4 is required for postnatal growth and maturation of the NMJ

Next, we tested whether membrane anchoring of LRP4 further enhances the function of the ECD or whether the ICD is required to achieve full functionality. To do this, we took advantage of a mouse line carrying a novel *Lrp4* knockin allele in which the ICD had been replaced with a Myc-tag (*Lrp4*$^{\Delta ICD/\Delta ICD}$, *Figure 1C*). Triangularis sterni muscle from these mice was isolated at P12 and NMJs were stained with anti-syntaxin antibodies and α-bungarotoxin. NMJs developed normally in the *Lrp4*$^{\Delta ICD/\Delta ICD}$ mice (*Figure 2B*, middle column), with synapse size indistinguishable from wild type (left column). These results are also consistent with the lack of an obvious neuromuscular phenotype in a strain of cattle affected by Mulefoot disease, where a naturally occurring splice site mutation in *Lrp4* results in truncation of the protein and a nearly complete loss of the cytoplasmic domain (*Johnson et al., 2006*).

By contrast, the impaired development of NMJs in *Lrp4*$^{ECD/ECD}$ mice continues during the postnatal period. AChR clusters were distributed more diffusely in P12, the triangularis sterni muscle of *Lrp4*$^{ECD/ECD}$

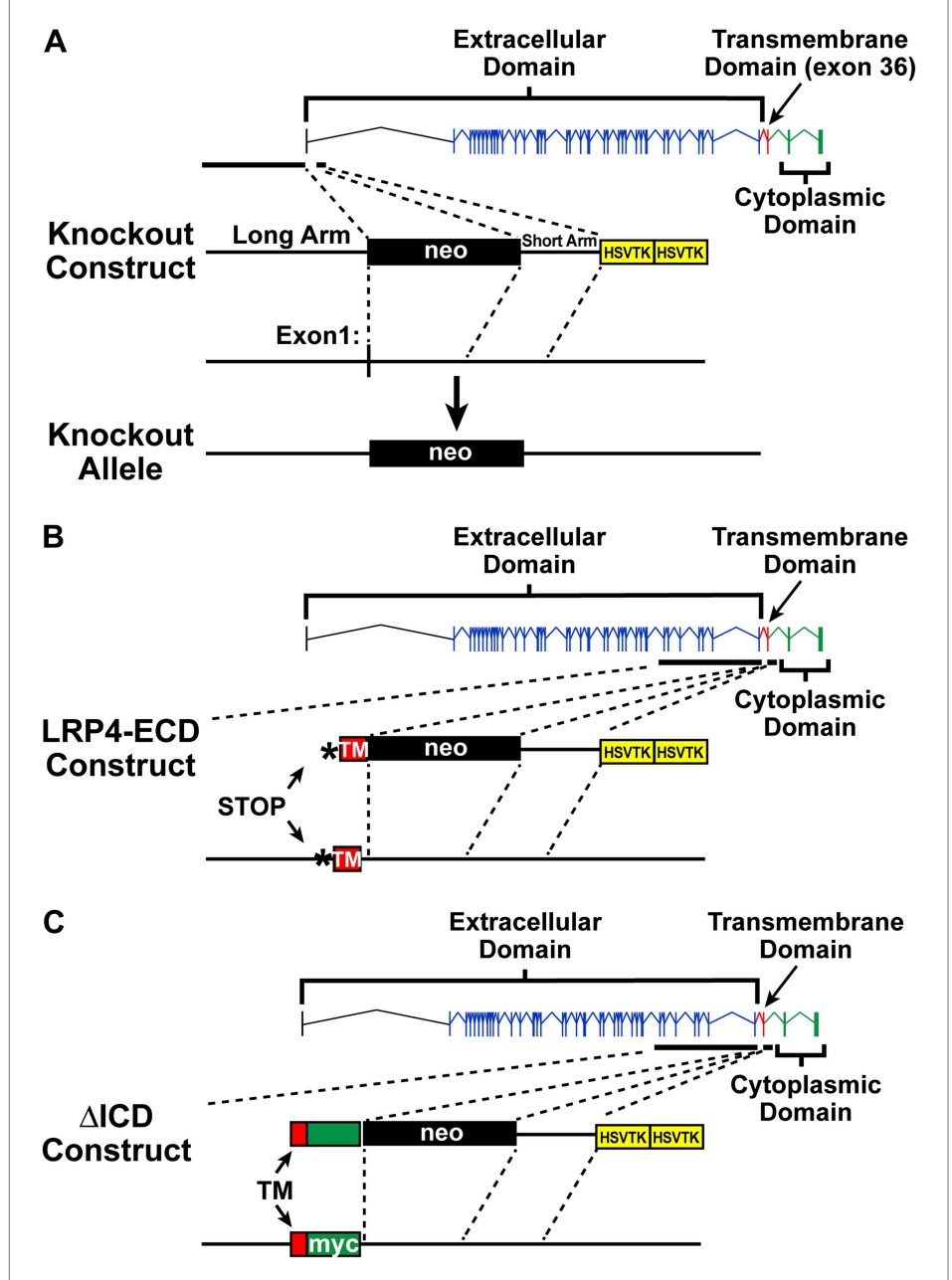

**Figure 1**. Targeting vectors. Diagrammatic representation of gene replacement strategies used to generate *Lrp4*[−/−] (null allele, **A**), *Lrp4*[ECD/ECD] (hypomorphic allele without membrane anchor, **B**), and *Lrp4*[ΔICD/ΔICD] (**C**) mice. Exons encoding the LRP4 ECD are depicted in blue and indicated by brackets. The exon encoding the transmembrane segment is shown in red, cytoplasmic domain encoding exons in green. To generate the *Lrp4* null allele, the transcription start site and exon 1 were replaced with a neo cassette (**A**). To generate the *Lrp4* ΔICD allele, a cDNA cassette encoding the transmembrane segment of LRP4 (TM, red) followed by a Myc epitope and bovine growth hormone 3'UTR was introduced into the targeting vector described by *Johnson et al. (2005)*. This results in the normal expression of the LRP4 ECD and transmembrane segment, but complete replacement of the ICD with a Myc tag (**C**). The generation of the LRP4 ECD allele has been described in *Johnson et al. (2005)* (**B**).

mice, compared with wild-type or *Lrp4*[ΔICD/ΔICD] mice (**Figure 2B**). In adult muscles, the NMJ from *Lrp4*[ECD/ECD] mice remained noticeably smaller than that from age-matched wild-type mice. As shown in **Figure 3A**, in the triangularis sterni muscle from 3-month old *Lrp4*[ECD/ECD] mice, the pre-synaptic nerve terminal and post-synaptic AChR clusters remained markedly smaller compared with wild type. By

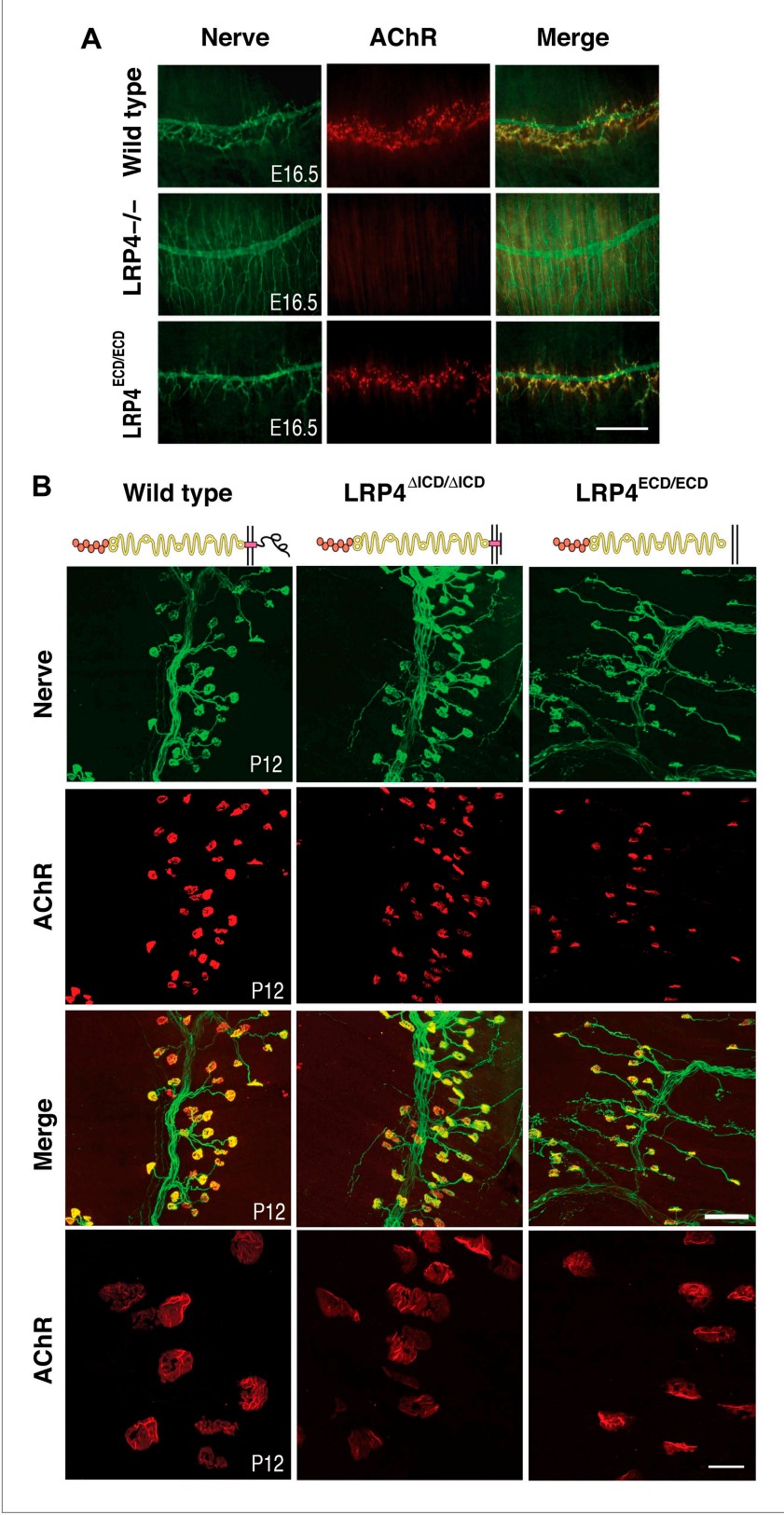

**Figure 2**. NMJ development in *Lrp4* mutant mice. (**A**) Embryonic diaphragm muscles (E16.5) from wild-type (top row), *Lrp4*⁻/⁻ (middle row) and *Lrp4*^ECD/ECD mice (bottom row) were double-labeled with anti-syntaxin antibodies (green) for decorating innervating fibers and with α-bungarotoxin (red) to detect AChRs. In both wild-type and *Figure 2. Continued on next page*

*Figure 2. Continued*

*Lrp4*<sup>ECD/ECD</sup> muscles, AChRs are clustered along the central region juxtaposed to nerve terminals. By contrast, AChR clusters are completely absent and NMJs fail to form in *Lrp4⁻/⁻* muscles. (**B**) Triangularis sterni muscles at postnatal day 12 were stained for nerves in green (anti-NF150 and anti-Syt2 antibodies) and AChRs in red (α-bungarotoxin). In both wild-type (left column) and *Lrp4*<sup>ΔICD/ΔICD</sup> mice (middle column), neuromuscular synapses were distributed within a narrow band near the nerve trunk. By contrast, neuromuscular synapses in *Lrp4*<sup>ECD/ECD</sup> mice (right column) were distributed across a broader region of the muscle, as prolonged nerve branches extended from the nerve trunk. Furthermore, AChR clusters are markedly smaller in the *Lrp4*<sup>ECD/ECD</sup> muscle, compared with the wild-type or *Lrp4*<sup>ΔICD/ΔICD</sup> muscle (bottom row: high-power views of AChR clusters). Scale bars, **A**: 300 µm; **B**: 200 µm (top three rows) and 50 µm (bottom row).

contrast, the NMJ in *Lrp4*<sup>ΔICD/ΔICD</sup> mice developed to similar size as wild type (***Figure 3B***). Thus, although the NMJ was established in *Lrp4*<sup>ECD/ECD</sup> embryos, it remained considerably smaller (compared with age-matched wild-type control) at postnatal and adult stages, indicating that LRP4 membrane anchoring is required for the growth and maturation of the NMJ. Intriguingly, the ultrastructure of the nerve terminal and post-synaptic membrane appeared largely normal—with abundant presynaptic vesicles and elaborate junctional folds (***Figure 3C***). We examined 55 nerve terminal profiles from 18 *Lrp4*<sup>ECD/ECD</sup> NMJs and 74 profiles from 23 control NMJs and found no statistical difference between control and *Lrp4*<sup>ECD/ECD</sup>.

Furthermore, despite a significant reduction in sizes, only a minor fraction (2%) of synapses was innervated by more than one axon in *Lrp4*<sup>ECD/ECD</sup> NMJs at P12, as was the case in the wild type. Moreover, the γ-ε switch also occurred normally in the *Lrp4*<sup>ECD/ECD</sup> mutant. By P16 γ-AChR subunit expression was no longer detectable on the protein or mRNA level in wild-type, *Lrp4*<sup>wt/ECD</sup> or *Lrp4*<sup>ECD/ECD</sup> muscle (data not shown). Therefore, deleting the transmembrane and intracellular domains of LRP4

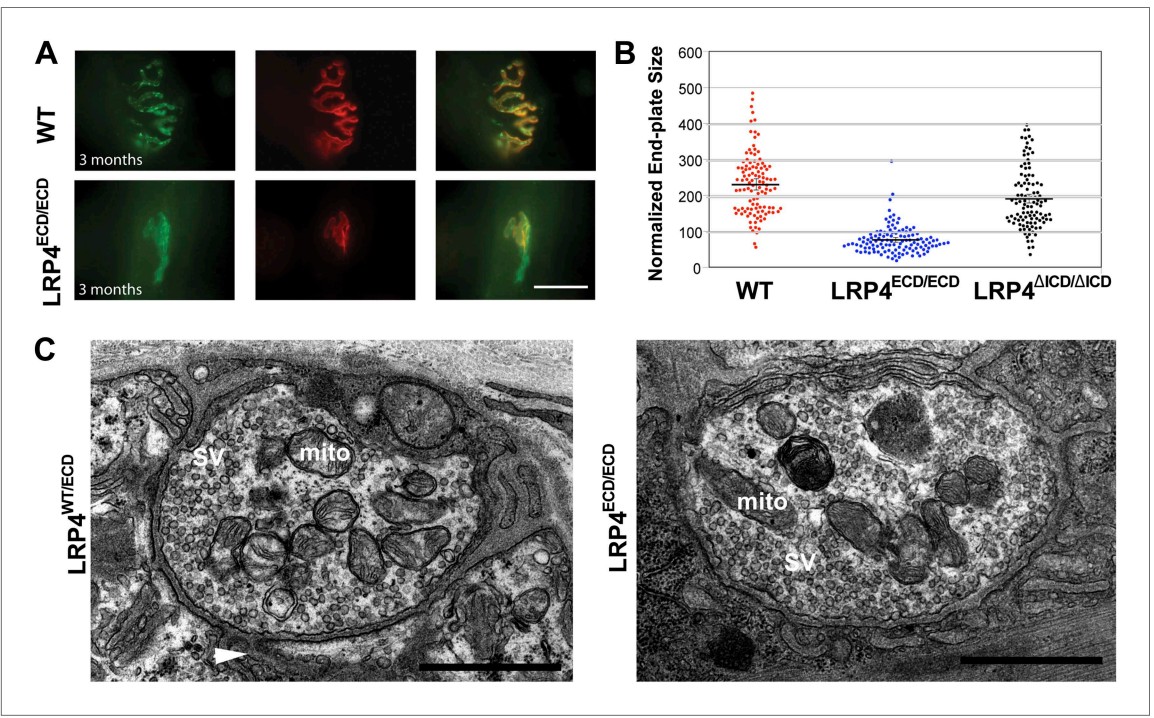

**Figure 3**. Reduced NMJ size in *Lrp4*<sup>ECD/ECD</sup> mice. (**A**) Triangularis sterni (TS) muscles from wild type (WT) and 3-month-old *Lrp4*<sup>ECD/ECD</sup> mice double-labeled with anti-syntaxin antibodies (green, nerve) and α-bungarotoxin (red, AChRs). The NMJ in the *Lrp4*<sup>ECD/ECD</sup> muscle is markedly smaller than in the wild-type muscle. (**B**) Size comparison of neuromuscular synapse areas in TS muscles in 3-month-old wild-type, *Lrp4*<sup>ECD/ECD</sup> and *Lrp4*<sup>ΔICD/ΔICD</sup> mice. (**C**) Electron micrographs of NMJs from lumbrical muscles in 2-month old *Lrp4*<sup>WT/ECD</sup> and littermate *Lrp4*<sup>ECD/ECD</sup> mice. Nerve terminals containing numerous synaptic vesicles and mitochondria are embedded in the muscle surface and junctional folds are apparent in both genotypes. Scale bars, 1 µm. 55 nerve terminal profiles from 18 *Lrp4*<sup>ECD/ECD</sup> NMJs and 74 profiles from 23 control NMJs were analyzed. Representative images are shown. Arrow indicates junctional fold. SV: synaptic vesicle; mito: mitochondria.

had no impact on the process of synapse elimination and AChR subunit switch (from the embryonic γ to the adult ε form) at the NMJ.

These findings indicate that the LRP4 intracellular domain harbors no essential elements for relaying the molecular signals that govern NMJ formation, but that membrane anchoring of LRP4 is required for maintaining the level of signal strength necessary for postnatal growth and maturation of the neuromuscular synapses. On the other hand, secretion of the extracellular domain of LRP4 into the pericellular space is apparently sufficient to initiate clustering of AChRs and thus manifestation of immature NMJs that can then be maintained at the prevailing reduced signal strength. This is consistent with the findings in conditional *Musk* knockout mice (*Hesser et al., 2006*), which showed NMJ disassembly in postnatal muscle upon conditional MuSK inactivation, indicating the requirement for continuous MuSK activity in postnatal muscle.

## Genetic interaction of *App* and *Lrp4* during NMJ development

This low level signaling may in part be mediated by the interaction of LRP4 ECD with MuSK (*Zhang et al., 2008*; *Kim et al., 2008b*). However, several ApoE receptors have also been reported to interact directly or indirectly with APP (*Kounnas et al., 1995*; *Ulery et al., 2000*; *Pietrzik et al., 2002*; *Andersen et al., 2005*; *Hoe et al., 2005*; *Pietrzik and Jaeger, 2008*; *Marzolo and Bu, 2009*), and APP itself has been shown to participate in NMJ development (*Wang et al., 2005*). Taken together, these findings suggested that LRP4 might act synergistically with APP to regulate NMJ development and maintenance. To test this hypothesis, we bred $Lrp4^{ECD/ECD}$ mice with $App^{-/-}$ mice to generate $Lrp4^{ECD/ECD}$ and *App* double mutant mice. We found that postnatal survival of compound mutant mice was markedly reduced when three wild-type *Lrp4* and *App* alleles were deleted (e.g., $Lrp4^{+/ECD}App^{-/-}$ or $Lrp4^{ECD/ECD}App^{+/-}$) and that the survival of double mutant mice ($Lrp4^{ECD/ECD}App^{-/-}$) was significantly reduced (p<0.01) (*Table 1*). Because of this accelerated loss of the compound double mutant animals before weaning, we analyzed the NMJ in E18.5 embryos rather than in adults, as normal Mendelian distribution was observed at E18.5 in these mice (data not shown). As has been reported earlier (*Wang et al., 2005*), normal size of AChR clusters was observed at the NMJ of $App^{-/-}$ embryos (*Figure 4A,B*, d–f, compared to wild type in subpanels a–c, *Figure 4C*). In $Lrp4^{ECD/ECD}$ embryos, AChR cluster size was similar to wild-type muscles, but AChR density was decreased (*Figure 4C*). However, in $Lrp4^{ECD/ECD}App^{-/-}$ double mutant embryos, both the density and the size of AChR clusters were significantly reduced compared to wild-type, $App^{-/-}$ or $Lrp4^{ECD/ECD}$ single mutant mice (*Figure 4C*). Furthermore, nerve terminal sprouting was markedly increased in the double mutant mice (*Figure 4A,B,j–l*). The significant reduction in the synaptic area of individual double mutant NMJs compared to wild type and $Lrp4^{ECD/ECD}$ is also apparent in the three-dimensional rendition of individual synapses (*Figure 4A,m–o*). Similar findings were obtained in $Lrp4^{ECD/ECD}$ mice lacking the amyloid precursor protein family member APLP2 (*Figure 5*), which cooperates with APP in the development of the NMJ (*Wang et al., 2005*).

## Biochemical interaction of APP and LRP4 extracellular domains

These results indicate that *Lrp4* and the *App* family members, *App* and *Aplp2*, interact genetically and functionally in the formation of NMJs and that loss of *App* or *Aplp2* greatly enhances the synaptic

**Table 1.** Postnatal survival of $Lrp4^{ECD/ECD}$;$App^{-/-}$ double mutant mice

| Genotype | Expected | Observed | Chi-square test |
|---|---|---|---|
| $Lrp4^{+/ECD}$;$App^{+/-}$ | 63 | 74 | n.s. |
| $Lrp4^{+/ECD}$;$App^{-/-}$ | 44 | 45 | n.s. |
| $Lrp4^{ECD/ECD}$;$App^{+/-}$ | 49 | 49 | n.s. |
| $Lrp4^{ECD/ECD}$;$App^{-/-}$ | 38 | 15 | * |

*Note*: 41% (20/49) of $Lrp4^{ECD/ECD}$;$App^{+/-}$ died within five months of age; 13% (6/45) of $Lrp4^{+/ECD}$;$App^{-/-}$ mice died within five months of age.

n.s., not significant.

*Significant at p<0.01. Survival was scored at weaning (P25). The 'Expected' column states the number of offspring of the indicated genotype according to the rules of Mendelian inheritance. The 'Observed' column states the number of offspring found to have the indicated genotype. Chi-square analysis of these data indicates whether the observed number significantly deviates from the expected number or not.

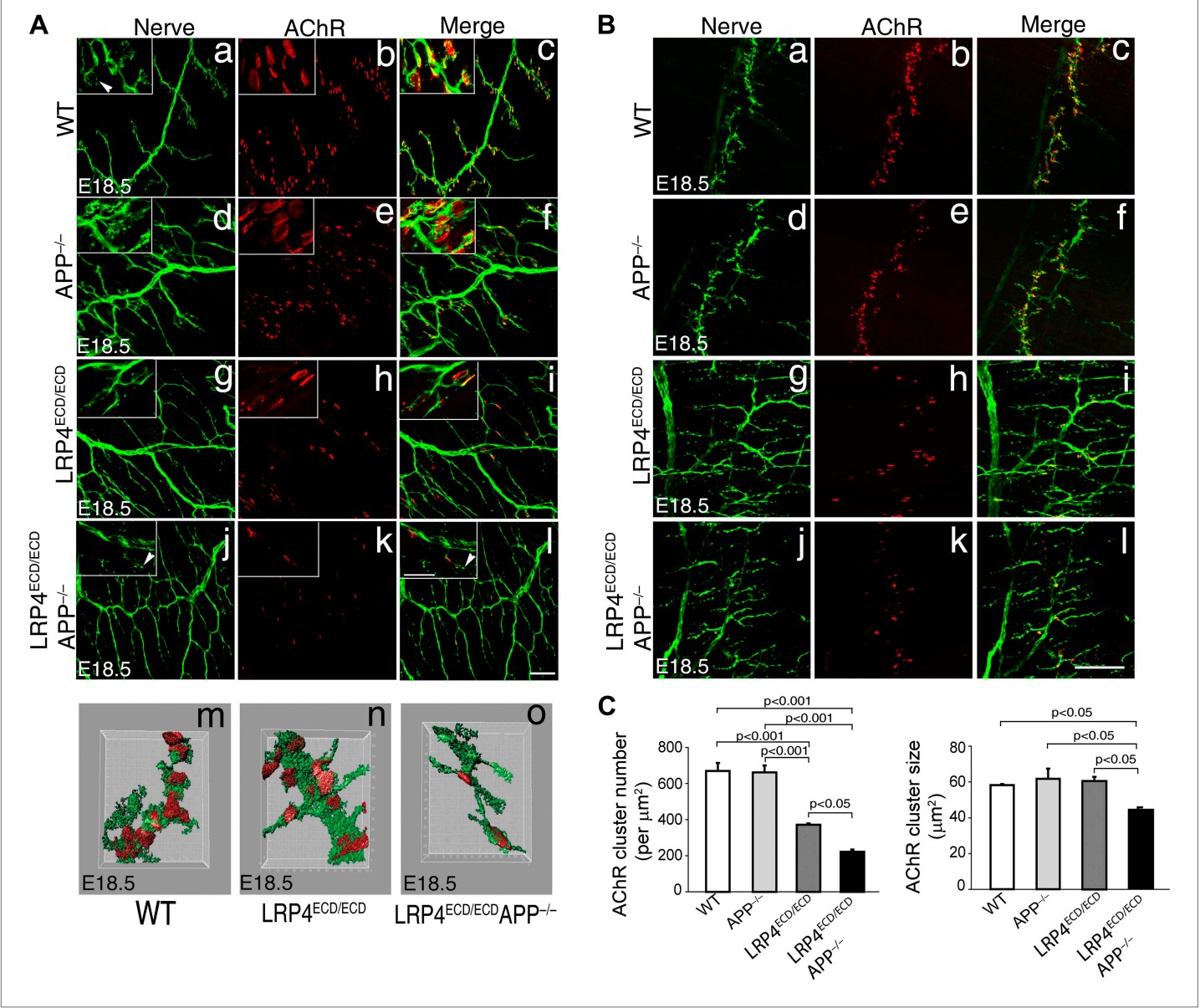

**Figure 4**. Impairment of pre-and post-synaptic development in *Lrp4*^ECD/ECD;*App*^−/− double mutant mice. (**A**) (**a–l**) Wholemount staining of TS muscles (E18.5) double-labeled with anti-neurofilament antibodies and anti-Syt2 antibodies (nerve, **a**, **d**, **g**, **j**) and α-bungarotoxin (AChR, **b**, **e**, **h**, **k**). Merged images are shown in panels **c**, **f**, **i** and **l**. The inset in each panel shows a high-power view of the image. AChR clusters were markedly reduced both in number and size in *Lrp4*^ECD/ECD;*App*^−/− mice (**j–l**), compared to wild type (**a–c**), *App*^−/− (**d–f**) or *Lrp4*^ECD/ECD (**g–i**). Furthermore, numerous terminal sprouts (arrowheads in the inset of **j** and **l**) were seen in *Lrp4*^ECD/ECD;*App*^−/− mutant mice, whereas the nerve terminals in the wild type (arrowhead in **a**) juxtaposed with postsynaptic AChR clusters. **m–o**: 3D reconstruction of confocal images of the NMJs in wholemount diaphragm muscles in wild type (**m**), *Lrp4*^ECD/ECD (**n**) and *Lrp4*^ECD/ECD;*App*^−/− (**o**) illustrate the reduced size of the NMJ in (**o**). Scale bars: **a–l**: 100 µm; inset: 20 µm. (**B**) Wholemount staining of diaphragm muscles (E18.5) double-labeled with anti-neurofilament antibodies and anti-Syt2 antibodies (nerve, **a**, **d**, **g**, **j**) and α-bungarotoxin (AChR, **b**, **e**, **h**, **k**). Low-power views of the left dorsal region of the diaphragm muscles. Scale bar, 400 µm. The number of AChR clusters was notably reduced in *Lrp4*^ECD/ECD mice, compared to wild-type and *App*^−/− mice. However both the size and number of AChR clusters were markedly reduced in *Lrp4*^ECD/ECD;*App*^−/− mice, compared to wild-type, *App*^−/− and *Lrp4*^ECD/ECD mice. (**C**) Quantitative analysis of AChR clusters size and numbers. The number of AChR clusters at the ventral regions of right hemi-diaphragm muscles (E18.5), normalized by muscle area (upper panel). Average size of AChR clusters (lower panel). The numbers of AChR clusters analyzed are: 144 wild type, 132 *App*^−/−, 131 *Lrp4*^ECD/ECD and 96 *Lrp4*^ECD/ECD;*App*^−/− (N = 3 mice for each genotype). Data are shown as average ± S.E.M. Pairwise multiple comparisons were carried out using Tukey's test and the statistical differences determined by one-way analysis of variance (ANOVA).

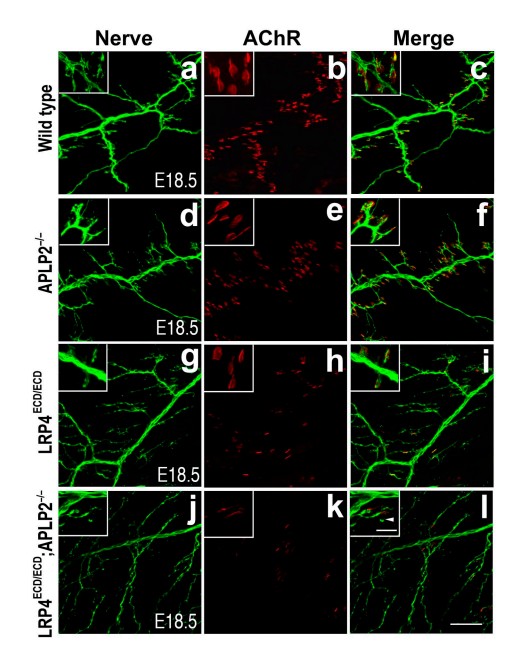

**Figure 5**. Impairment of pre-and post-synaptic development in *Lrp4*ECD/ECD;*Aplp2*–/– double mutant mice. Wholemount staining of triangularis sterni muscles (E18.5) double-labeled with anti-neurofilament antibodies and anti-Syt2 antibodies (nerve, **a**, **d**, **g**, **j**) and α-bungarotoxin (AChR, **b**, **e**, **h**, **k**). Merged images are shown in panels **c**, **f**, **i** and **l**. The inset in each panel shows a high-power view of the image. The nerve terminals and AChR clusters were markedly reduced (both in number and size) in *Lrp4*ECD/ECD;*Aplp2*–/– mice (**j**–**l**), compared to wild type (**a**–**c**), *Aplp2*–/– (**d**–**f**) or *Lrp4*ECD/ECD (**g**–**i**). Arrowhead in the inset of **l** indicates a nerve terminal sprout. Scale bars: **a**–**l**: 100 µm; inset: 20 µm.

defect in *Lrp4*ECD/ECD mice. To test, whether APP and LRP4 interact directly, analogous to the interactions of APP with LRP1 (***Kounnas et al., 1995***) and Apoer2 (***Hoe et al., 2005***), we performed a binding analysis using recombinant APP and LRP4 fusion proteins. The LRP4 ligand-binding domain (LBD) was fused to maltose-binding protein (MBP) and the APP ECD was fused to the constant Fc region of human IgG. ***Figure 6A*** shows that specific, ectodomain-dependent interaction of both fusion proteins does indeed occur. To further corroborate these findings, we performed a cellular aggregation analysis in which we transiently co-transfected 293 cells with GFP and full length APP, or RFP and full length LRP4 plasmids, respectively. Equal numbers of red, green or both types of fluorescent cells were mixed, and the number of aggregates was scored (***Figure 6B***). Significantly more and larger aggregates formed in the presence of APP and LRP4 than in control experiments in which cells were only transfected with GFP, RFP, APP+GFP, and LRP4+RFP, respectively (***Figure 6C***, note approximately equal numbers of APP and LRP4 expressing cells in the aggregates in ***Figure 6B***). Although homophilic interaction of APP has been reported (***Soba et al., 2005***), no increase in the number of large clusters was detected in wells containing only APP or LRP4 expressing cells, indicating a higher affinity of APP for LRP4 than for itself.

## APP and agrin cooperate to promote AChR clustering

The direct interaction between the LRP4 LBD and APP ectodomain suggested that pre- and/or post-synaptically expressed APP itself might be capable of engaging LRP4, thereby inducing MuSK phosphorylation and thus triggering AChR clustering on its own or cooperatively with the neural form of agrin (henceforth referred to as agrin unless stated otherwise). To test for this possibility, we exposed isolated myotubes to recombinant dimeric APP-Fc or RAP-Fc, an ER chaperone that binds the extracellular domains of ApoE receptors, that is LDL receptor family members (***Fisher et al., 2006***) including LRP4 (***Ohazama et al., 2008***), with high affinity and thereby prevents the binding of most cognate ligands to this class of receptors (***Herz et al., 1991***). To facilitate MuSK ligation, the Fc fusion proteins were immobilized on protein A beads and beads were incubated with isolated myotubes. ***Figure 7A*** shows that APP-Fc beads, but not RAP-Fc beads, significantly increased AChR clustering in the absence of exogenous neuronal agrin. When the same experiment was performed in the presence of a low concentration of agrin (0.1 ng/ml), again only APP-Fc, but not RAP-Fc, significantly increased AChR clustering. These results indicate that APP and agrin function synergistically through LRP4 to stimulate AChR clustering and NMJ formation. Interestingly, soluble APP-Fc not bound to beads was also able to promote AChR clustering, suggesting that focal clustering of APP on opposing beads is not required, but that the interaction of APP, either in a dimeric (APP-Fc) or monomeric (APP-FLAG) form (***Figure 7B***), with the myotube surface is sufficient to induce initial clustering. The soluble LRP4-ligand binding domain (MBP-LRP4) was also capable of inducing clustering, alone or together with the APP-ectodomain (***Figure 7B***). Moreover, the failure of RAP-Fc to induce AChR clustering suggests that LRP4 dimerization alone is not sufficient, but that possibly the formation of an extracellular scaffold (***Bromann et al., 2004***) through interactions with other surface proteins, which can be mediated by APP and LRP4 ECD, is required.

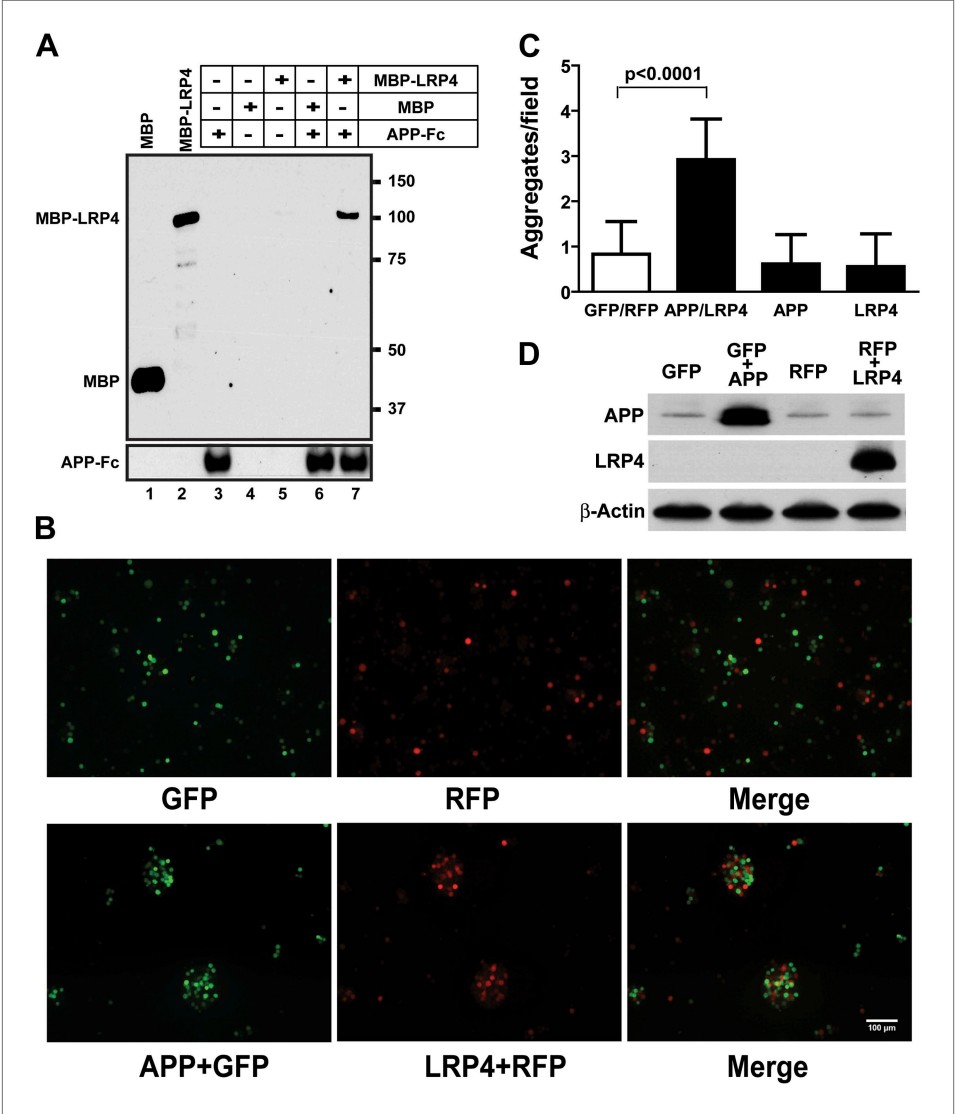

**Figure 6**. APP interacts with LRP4. (**A**) APP-Fc-bead conjugates or blank beads were incubated with 100 ng purified MBP or MBP-LRP4LBD for 4 hr at 4°C prior to immunoprecipitation with anti-MBP followed by immunoblotting with anti-hFc antibody (lane 3–7). Lanes 1 and 2 show MBP (control protein) and MBP-LRP4LBD, respectively. (**B**) GFP- and RFP-expressing HEK293T or APP/GFP- and LRP4/RFP-expressing HEK293T cells were mixed in suspension and incubated for 30 min at 37°C before capturing cell aggregates under a fluorescent microscope. (**C**) The number of aggregates bigger than 3000 μm² from twelve randomly selected images was determined. A significantly (p<0.0001) increased number of aggregates were present only when APP and LRP4 expressing cells were mixed. (**D**) APP and LRP4 protein levels in the transfected HEK293T cells.

## APP induces MuSK phosphorylation independent of agrin

These findings were further corroborated by determining the effect of APP-Fc on MuSK phosphorylation in the absence or presence of different concentrations of agrin (*Figure 7C*). At low concentrations (0.1 nM) agrin alone increased MuSK phosphorylation by 5.4-fold over baseline. RAP-Fc was used as a control and had no effect, whereas APP-Fc alone also dose-dependently increased MuSK phosphorylation, although not as effectively as agrin (2.4, 3.2 and 4.3-fold over baseline at 1×, 10× or 100× molar concentrations, respectively, compared to 0.1 nM agrin alone). However, at low (0.1 nM) to intermediate (1 nM) concentrations APP-Fc significantly increased MuSK phosphorylation by agrin further (*Figure 7C,D*). At higher concentration (10 nM), APP-Fc became less effective, suggesting that APP and agrin cooperatively enhance MuSK phosphorylation within a narrow physiological concentration range.

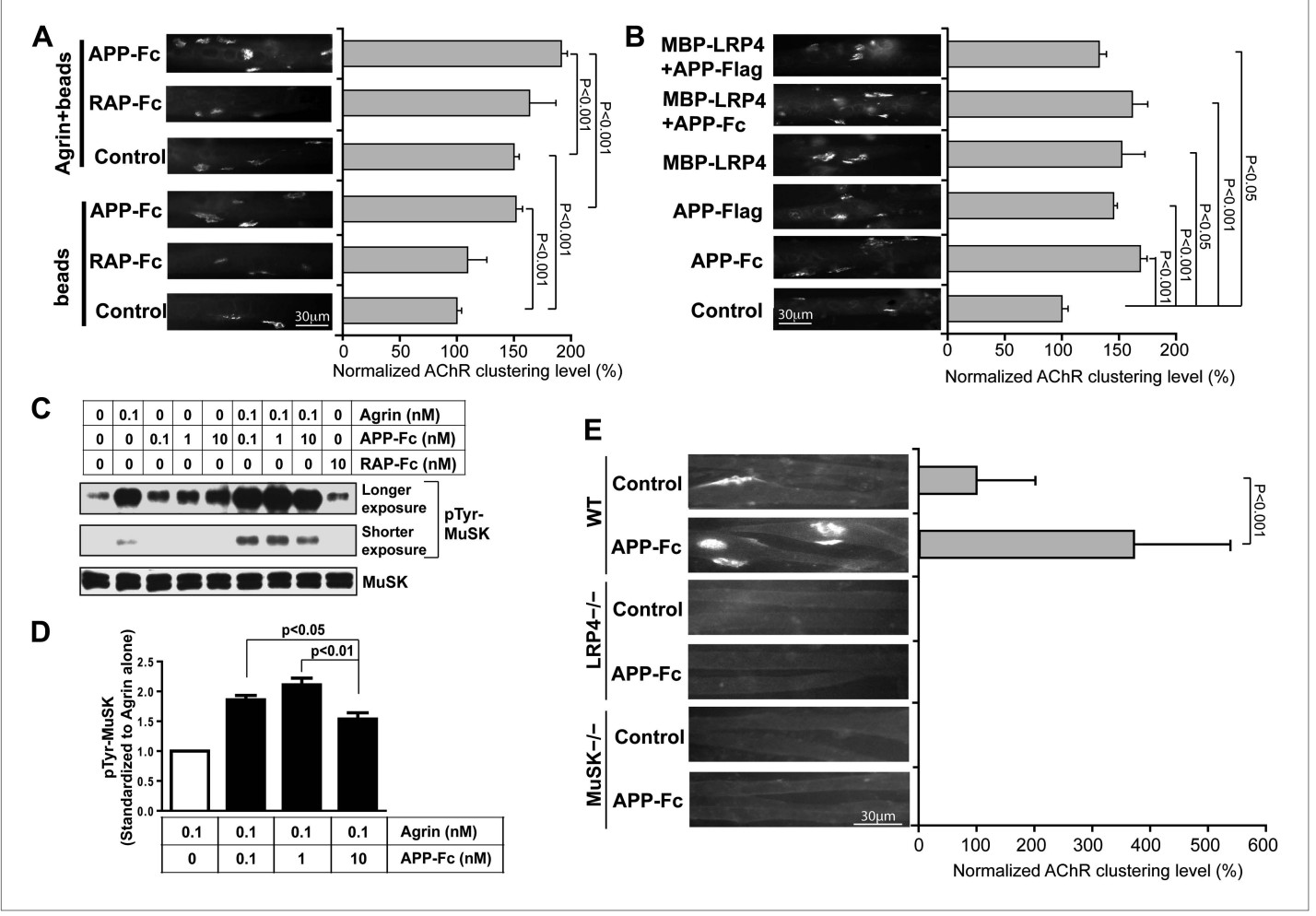

**Figure 7**. APP activates MuSK and promotes AChR clustering. (**A**) C2C12 myotubes were incubated with the indicated beads preloaded with APP-Fc or RAP-Fc, respectively, in the absence or presence of 0.1 ng/ml agrin for 24 hr prior to labeling AChRs. The number of average AChR clusters per 200 µm myotube was counted from 50 randomly captured images and normalized to control levels. (**B**) Monomeric and divalent recombinant APP fusion proteins and MBP-LRP4, alone or in combination with APP fusion proteins, are equally effective in inducing AChR clustering. C2C12 myotubes were incubated with 1 µg/ml of soluble monovalent APP (APP-Flag), divalent APP (APP-Fc), or MBP-LRP4, alone or in combination with APP fusion proteins, for 24 hr prior to labeling AChRs. (**C**) Myotubes were incubated for 30 min at 37°C with the indicated relative molar concentrations of proteins prior to immunoblotting for total MuSK and tyrosine phosphorylated MuSK (pTyr-MuSK). 0.1 nM Agrin = 9 ng/ml, 0.1 nM APP-Fc = 13 ng/ml, 10 nM RAP-Fc = 720 ng/ml. (**D**) Relative amounts of pTyr-MuSK were quantified from three independent experiments. At all APP concentrations MuSK phosphorylation was significantly increased for agrin alone (0.1 nM APP, p=0.0003; 1 nM APP, p=0.0002; 10 nM APP, p=0.0031). (**E**) APP-induced AChR clustering requires LRP4 and MuSK. Primary myotubes cultured from wild-type (WT), *Lrp4−/−*, and *Musk−/−* mouse embryos (E18.5) were incubated in the absence (control) or presence of APP-Fc for 24 hr prior to labeling AChRs. Statistical analysis by Student's *t*-test. Scale bars, 30 µm.

As was expected from the analysis of mice genetically deficient for LRP4 or MuSK, respectively, recombinant APP-Fc failed to induce any detectable AChR clustering in LRP4 or MuSK deficient myotubes (*Figure 7E*), indicating that APP can enhance but not bypass the essential activities of LRP4 and MuSK in NMJ formation.

## APP, LRP4, and agrin interact cooperatively

These observations suggested that APP itself might interact directly with agrin and that this interaction might serve to cooperatively enhance a hetero-oligomeric interaction between APP, LRP4 and agrin to maximize MuSK phosphorylation and thereby focus AChR clustering to the narrow patch of the membrane destined to harbor the emerging NMJ. To test this hypothesis, we determined whether APP, LRP4 and MuSK form a stable complex in muscle in vivo. Wild-type embryonic muscle proteins were precipitated with antibodies against LRP4, as well as Apoer2 and LRP1, two other LDL receptor family

members that are expressed in muscle and are known to interact with APP. Remarkably, only LRP4 co-precipitated with APP and with MuSK, indicating that these three proteins are already present in muscle in a preformed stable complex (*Figure 8A*).

Next, we performed a qualitative interaction assay to determine whether agrin can interact independently with the APP ectodomain. To this end, LRP4-Fc, APP-Fc and RAP-Fc fusion proteins were prepared and incubated with a fixed concentration of agrin (100 ng/ml). LRP4-Fc efficiently bound agrin at much lower concentrations than was required to achieve comparable binding to APP-Fc (*Figure 8B*), while RAP-Fc did not interact with agrin at all. To test whether the ability of APP to bind agrin, albeit at much lower affinity than LPR4, translates into a cooperative enhancement of the interaction of APP with LRP4, we incubated a fixed concentration of APP-Fc (100 ng/ml) with a fixed amount of LRP4-MBP (100 ng/ml) in the absence or presence of increasing amounts of agrin. As shown in *Figure 8C*, agrin dose-dependently enhanced the interaction of LRP4 with APP at 10 ng/ml (1.3-fold) up to an optimal concentration ratio of 30 ng/ml (threefold higher binding than in the absence of agrin). Higher concentrations of agrin potently inhibited the interaction of LRP4 with APP

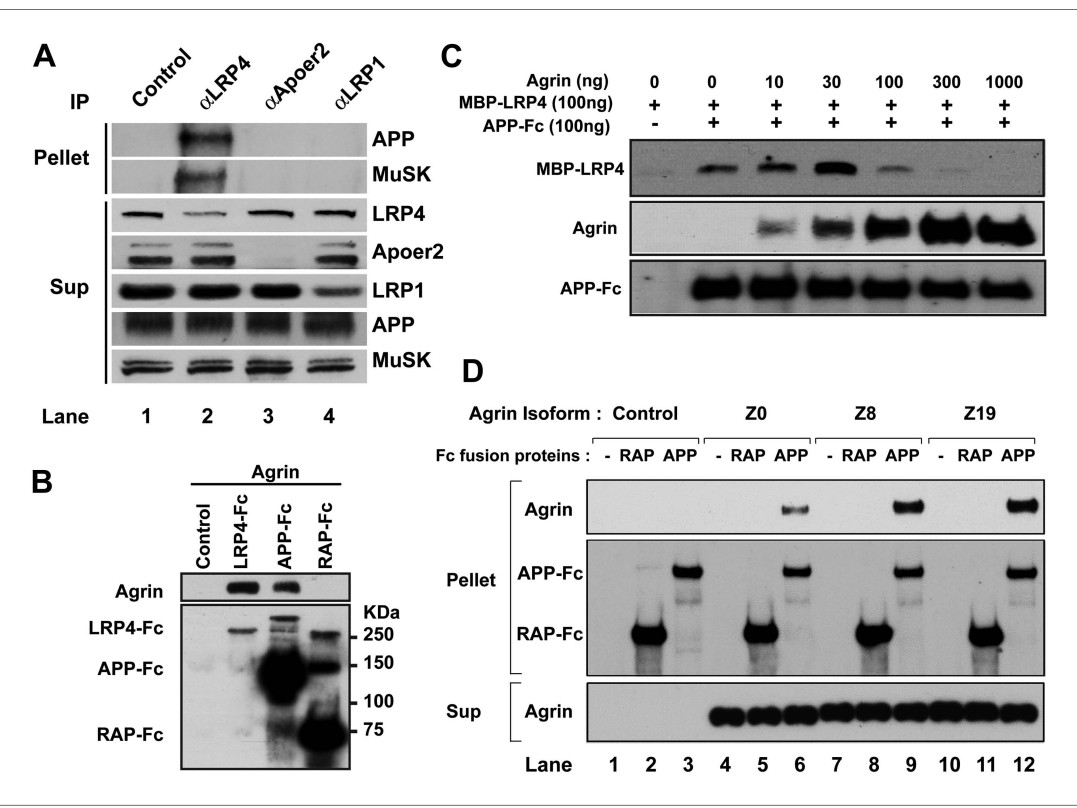

**Figure 8**. Interactions between agrin, APP and LRP4. (**A**) APP, LRP4 and MuSK form a stable complex in muscle in vivo. Proteins were extracted from wild-type embryo (E14.5) muscles and incubated with polyclonal anti-LRP4, anti-Apoer2, or anti-LRP1 antibodies overnight at 4°C followed by adsorption to Protein A Dynabeads. Supernatants were efficiently immunodepleted of LRP4, Apoer2 and LRP1 with the respective antibodies. APP and MuSK coprecipitate with LRP4, but not with the other LRP members Apoer2 and LRP1, indicating the presence of a physiological complex consisting of LRP4, APP and MuSK in muscle in vivo. (**B**) Control (non-transfected), LRP4-Fc, APP-Fc, or RAP-Fc containing HEK293T supernatants were adsorbed with Protein A sepharose beads. Conjugated beads were incubated with 1 µg neural agrin in a final volume of 1 ml for 4 hr at 4°C, followed by immunoblotting for bound Fc proteins and agrin. (**C**) APP-Fc-bead conjugates or blank beads were incubated with purified 100 ng of MBP-LRP4LBD and 0–1000 ng of neural agrin for 4 hr at 4°C. The precipitate was analyzed by immunoblotting with anti-MBP, anti-agrin, and anti-Fc antibodies to detect cooperative protein interactions among LRP4, APP, and agrin. (**D**) Control (lanes 1–3), agrin Z0 (lanes 4–6), agrin Z8 (lanes 7–9), or agrin Z19 (lanes 10–12) containing culture supernatants were incubated overnight at 4°C in the absence (lanes 1, 4, 7, 10) or presence of RAP-Fc (lanes 2, 5, 8, 11) or APP-Fc (lanes 3, 6, 9, 12) prior to adsorption to Protein A coupled Dynabeads and magnetic isolation. Bound agrin isoforms and Fc proteins were determined by immunoblotting with anti-Myc and anti-Fc antibodies.

to 60% of baseline at 100 ng/ml and 10% at 300 ng/ml, respectively. At 1000 ng/ml, which equates to an approximately 15-fold molar excess of agrin, LRP4 binding to APP was completely abolished. This finding suggests that LRP4 and APP may bind agrin through different epitopes. Independent binding of different agrin molecules to LRP4 and APP may thus induce steric inhibition, which prevents the formation of the favored hetero-oligomeric complex consisting of LRP4, APP, agrin and, on the surface of the myotube, MuSK.

Agrin occurs in multiple splice forms (*McMahan et al., 1992*; *Ferns et al., 1992*, *1993*; *Burgess et al., 2000*) of which the Z0 form is expressed by non-neural tissues such as muscle, while Z8 and Z19 are neural-specific. To test whether APP selectively interacts with specific agrin isoforms, we performed pull-down analysis with APP-Fc. *Figure 8D* shows that all three agrin isoforms (Z0, Z8 and Z19) interact with APP, although Z0 binding to APP appears less robust than Z8 or Z19 does.

## AChR prepatterning is maintained in *App/Aplp2* double mutant muscle

To determine whether muscle prepatterning is affected in $App^{-/-}$;$Aplp2^{-/-}$ double mutant mice, we analyzed E14.5 muscles. Consistent with previous report that AChRs clusters are prepatterned in E14.5 muscle (*Lin et al., 2001*, *Chen et al., 2011*), AChR clusters were aligned along the central region of E14.5 wild-type muscle, and the majority of AChR clusters were near, but not directly apposed by presynaptic nerve terminals (*Figure 9*). Similarly, prepatterned AChRs were distributed along the central region of $App^{-/-}$;$Aplp2^{-/-}$ muscle, although individual AChR clusters appear less robust compared with those in the wild-type, $App^{-/-}$ or $Aplp2^{-/-}$ muscles. These results demonstrate that pre-patterned AChR clustering is maintained in the absence of APP and APLP2.

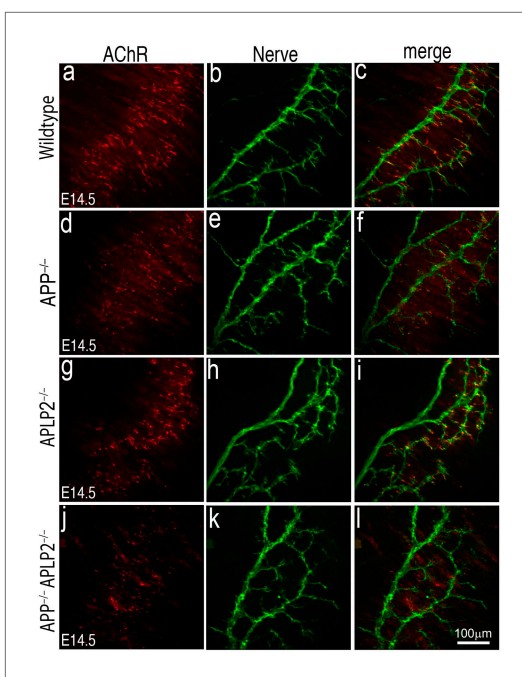

**Figure 9**. AChR prepatterning in the absence of APP and APLP2. Wholemounts of embryonic diaphragm muscles (E14.5) from wild type (**a–c**), $App^{-/-}$ (**d–f**), $Aplp2^{-/-}$ (**g–i**) and $App^{-/-}$;$Aplp2^{-/-}$ (**j–l**) embryos were double-labeled with α-bungarotoxin (red) and anti-syntaxin antibodies (green). In wild-type and all mutant embryos, AChR clusters were confined to the central regions of diaphragm muscles indicating AChR prepatterning is independent of APP and APLP2. However, individual AChR clusters appear less robust in $App^{-/-}$;$Aplp2^{-/-}$ muscle compared with those in the wild-type, $App^{-/-}$ or $Aplp2^{-/-}$ muscles. Scale bar: 100 μm.

## Localization of LRP4 in the absence of APP and APLP2

We next asked if removing membrane anchoring of LRP4 (as in $Lrp4^{ECD/ECD}$ mice) or deleting *App* and *Aplp2* (as in *App/Aplp2* double mutant mice) might affect LRP4 protein distribution in muscles. We generated anti-LRP4 antibodies against the extracellular domain of LRP4 and performed immunofluorescence staining on embryonic muscles. As expected, no LRP4 labeling was detected in $Lrp4^{-/-}$ muscle (*Figure 10A,k*; *Figure 10C,b*). By contrast, specific LRP4 staining was clearly visible at the NMJ in E18.5 WT muscle (*Figure 10A,h*), although the staining was markedly weaker at earlier developmental stages (E14.5 [*Figure 10A,b*] and E16.5 [*Figure 10A,e*]). Similarly, APP was also localized at the NMJ in E18.5 WT muscle (*Figure 10B*). In $Lrp4^{ECD/ECD}$ muscle (*Figure 10C,e*), we detected LRP4 staining at the NMJ, but at reduced levels compared with WT muscle (*Figure 10D*). Strikingly, LRP4 staining was also reduced in $App^{-/-}$;$Aplp2^{-/-}$ double mutant muscle (*Figure 10C,h*), compared to WT muscle (*Figure 10D*). In adult muscle (hindlimb, 10-week-old), higher LRP4 protein (*Figure 10E*) and mRNA (*Figure 10F*) expression were detected in $Lrp4^{ECD/ECD}$ compared to WT muscle. Taken together, these data indicate a decreased LRP4 level at the NMJ in $Lrp4^{ECD/ECD}$ and $App^{-/-}$;$Aplp2^{-/-}$ embryonic muscles.

## Discussion

Our results have revealed previously unrecognized interactions at the NMJ between LRP4 and

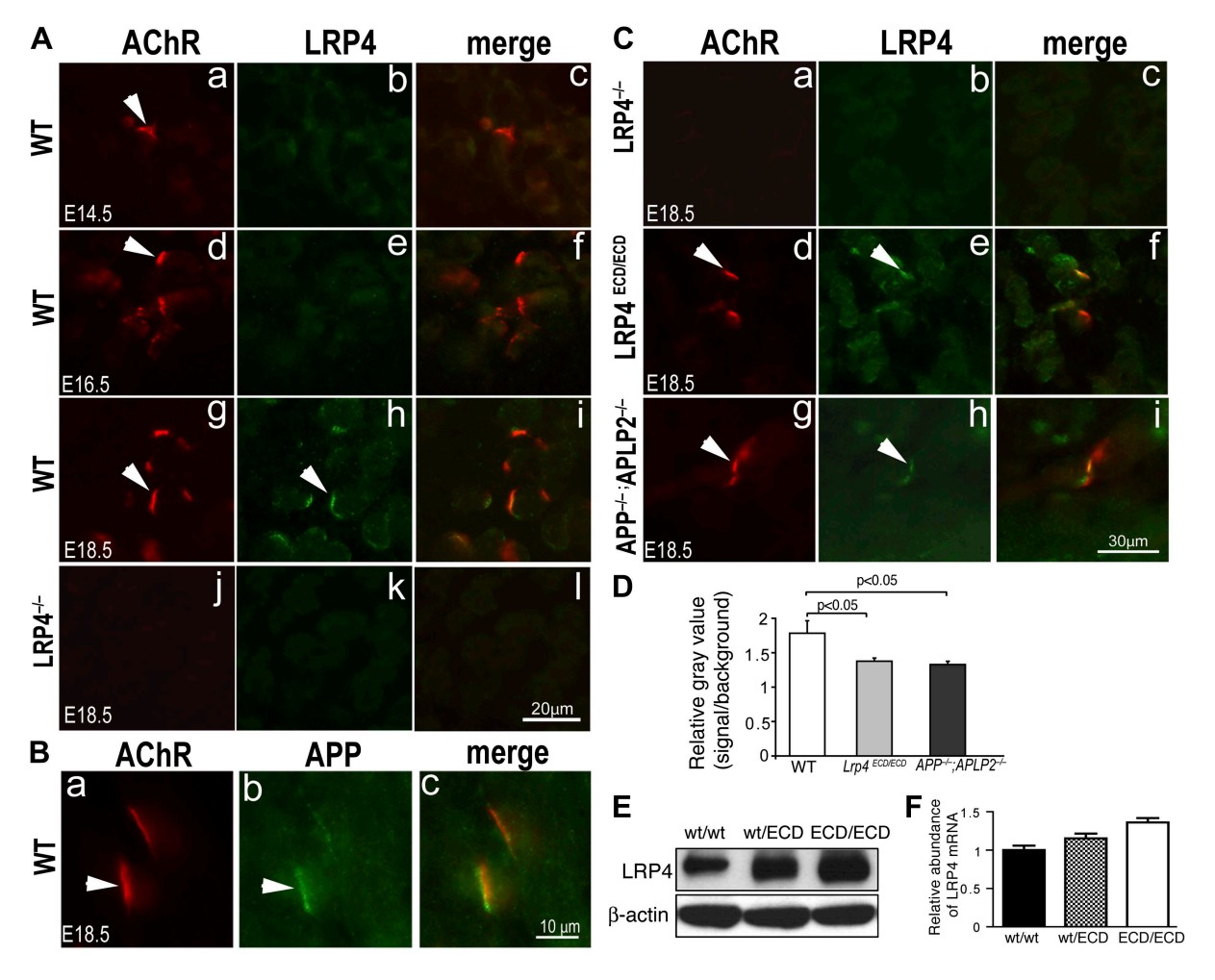

**Figure 10**. LRP4 localization in *Lrp4*[ECD/ECD] and *App*[−/−];*Aplp2*[−/−] mutant mice. (**A**) Cross sections of hindlimb muscle from E14.5 (**a**–**c**), E16.5 (**d**–**f**) and E18.5 (**g**–**i**) wild type (WT) and E18.5 *Lrp4*[−/−] (**j**–**l**) mice were double-labeled with anti-LRP4 antibodies (green) and Texas-Red conjugated α-bungarotoxin (red), which marks the site of the NMJ (arrowhead in **a**, **d**, **g**). LRP4 staining appears at low level in E14.5 WT and E16.5 WT muscle (compared with *Lrp4*[−/−] muscle), and become highly concentrated at the NMJ in E18.5 WT muscle (arrowhead in **h**). Scale bar in **A**: 20 µm. (**B**) Localization of APP at the NMJ. Wholemount diaphragm muscle from E18.5 WT mice was double-labeled with anti-APP antibodies (**b**) and Texas-Red conjugated α-bungarotoxin (**a**). APP (arrowhead in **b**) is localized at the NMJ (arrowhead in **a**), as shown in the merged image in **c**. Scale bar in **B**: 10 µm. (**C**) Cross sections of hindlimb muscles from *Lrp4*[−/−] (**a**–**c**), *Lrp4*[ECD/ECD] (**d**–**f**) and *App*[−/−];*Aplp2*[−/−] (**g**–**i**) mice (E18.5) were double-labeled with anti-LRP4 antibodies (green) and α-bungarotoxin (red). LRP4 was detected at the NMJ in *Lrp4*[ECD/ECD] (arrowhead in **e**) and *App*[−/−];*Aplp2*[−/−] (arrowhead in **h**), but not in *Lrp4*[−/−] muscle (**b**). LRP4 expression appeared diffused in *Lrp4*[ECD/ECD] and in *App*[−/−];*Aplp2*[−/−] muscles compared with age-matched WT muscle (see **A**). Scale bar in **C**: 30 µm. (**D**) Quantification of relative fluorescence intensity for LRP4 immunostaining on hindlimb myofibers from WT, *Lrp4*[ECD/ECD] and *App*[−/−];*Aplp2*[−/−] mice (E18.5). Gray value of LRP4 staining at the NMJ (defined as signal) and within the sarcoplasm (defined as background) of the same myofiber was separately measured using ImageJ. The ratio of signal to background was then calculated for each individual myofiber. The bar graph shows significant ($p < 0.05$) decreases in LRP4 staining in *Lrp4*[ECD/ECD] ($1.38 \pm 0.05$, n = 23 myofibers) and *App*[−/−];*Aplp2*[−/−] ($1.33 \pm 0.05$, n = 22 myofibers), compared with WT muscle ($1.78 \pm 0.18$, n = 19 myofibers). (**E**) Anti-LRP4 and beta-actin (loading control) immunoblot shows increased level of LRP4 protein expression in *Lrp4*[ECD/ECD] and *Lrp4*[WT/ECD] muscle, compared with WT muscle (from the hindlimb muscles of 10-week-old mice). (**F**) *Lrp4* mRNA expression in hindlimb muscles (10-week-old mice) was determined by quantitative PCR. Levels of *Lrp4* mRNA were normalized to cyclophilin mRNA levels. Results are shown as average ± SD of triplicates.

APP on the one hand and APP and agrin on the other. Genetic epistasis experiments show that LRP4 and APP functionally interact to regulate NMJ development and maintenance. APP may accomplish this through several independent mechanisms: first, APP interacts directly with LRP4 and thereby is capable of activating MuSK (at a low level and independent of agrin) by ligating and clustering LRP4. Second, APP can also directly bind to agrin, albeit at lower affinity than LRP4. However, agrin

cooperatively increases the stoichiometric interaction of APP with LRP4, which may serve to focus MuSK activation and thereby prevent aberrant AChR clustering. Third, APP and its homologues, which are expressed pre- and post-synaptically (*Wang et al., 2005*), bind homomerically and hetero-merically to each other (*Soba et al., 2005*), thereby generating or strengthening attachment sites for nerve terminals (*Torroja et al., 1999*; *Akaaboune et al., 2000*; *Soba et al., 2005*; *Wang et al., 2005*). By interacting simultaneously with LRP4 and agrin, APP and potentially APP-like proteins such as APLP2, which also participates in NMJ formation (*Wang et al., 2005*), would further serve to recruit all components necessary for AChR clustering into an area of maximal density, thereby ensuring that AChR clustering is restricted in vivo to the site of nerve contact. A rendition of this model is shown in *Figure 11A*.

We have shown that LRP4 ECD is sufficient to initiate NMJ formation at embryonic stages but not sufficient for postnatal maturation, which requires a progressive increase of MuSK activation. The secreted LRP4 ectodomain that is encoded by our LRP4 ECD allele is presumably retained at the cell surface by simultaneous interactions with APP (possibly also APLPs), MuSK and agrin, but at levels that are insufficient to sustain the signal strength required for postnatal expansion of the synapse (*Figure 11B*). Also supporting this model is the finding that initial AChR clustering is independent of

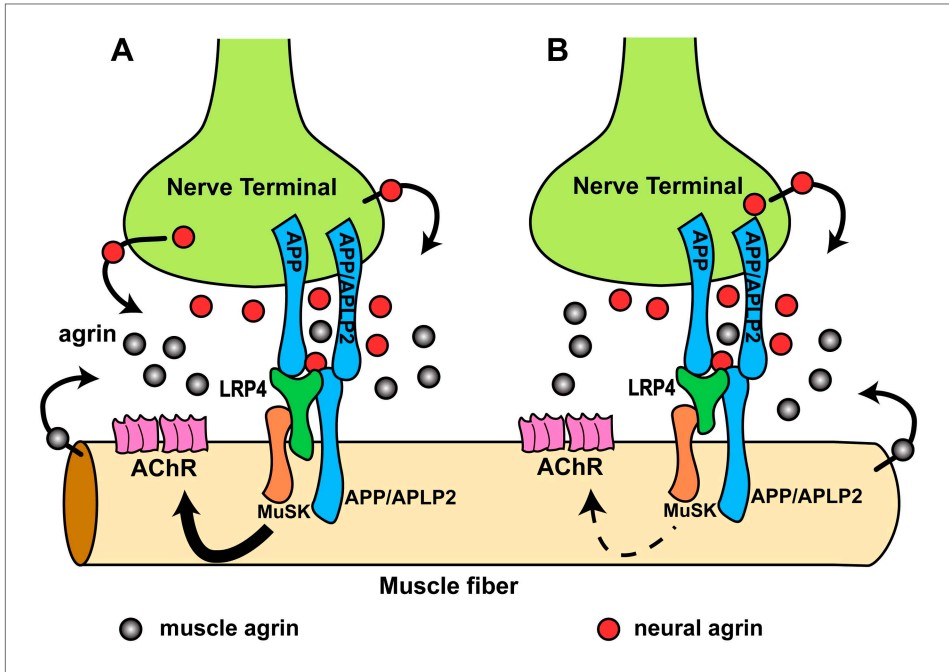

**Figure 11**. Hypothetical model of the interactions of APP/APLP2, LRP4, agrin and MuSK during NMJ formation. (**A**) Our genetic, biochemical and functional results suggest that APP interacts with LRP4 and agrin to regulate NMJ formation. APP–LRP4 interaction in the absence of agrin may be able to activate MuSK signaling (*Figure 7C*) at low levels to promote AChR clustering on the muscle fiber membrane, raising the possibility that nerve and muscle-derived APP and APLP2 homo- or heterodimerization may be able to cooperatively promote AChR clustering in the absence of agrin, which provides a potential mechanism for the earlier observation that initial AChR clustering is independent of agrin (*Lin et al., 2001*; *Yang et al., 2001*). Neural agrin (red) enhances the APP–LRP4 interaction and strongly activates MuSK (*Figures 7C and 8C*), which is consistent with the role of agrin in the stabilization and maintenance of NMJs. Cooperative binding of APP and agrin to LRP4 (*Figure 8C*) would further strengthen LRP4–APP interaction on muscles, promoting synaptic differentiation and postnatal maintenance of NMJs (solid line with arrow in muscle fiber cytoplasm). By contrast, muscle agrin (gray), which cannot activate MuSK, would compete with decreasing concentrations of diffusible neural agrin at increasing distance from the NMJ and thus keep AChR clustering outside the nerve contact site suppressed. (**B**) Loss of the LRP4 membrane anchor results in a secreted ectodomain that remains partially functional through bivalent interactions with MuSK (*Zhang et al., 2008*; *Kim et al., 2008b*) and APP, resulting in reduced signaling (dashed line with arrow) and thus a hypomorphic NMJ developmental defect compatible with embryonic NMJ formation, but impaired postnatal maturation and maintenance of the NMJ.

neuronal agrin (*Lin et al., 2001*; *Yang et al., 2001*) and that rudimentary NMJs form in agrin-deficient mice (*Gautam et al., 1996*), which suggested the existence of a second nerve derived organizing signal. By contrast, NMJs completely fail to form in *Musk* (*DeChiara et al., 1996*) and *Lrp4* (*Weatherbee et al., 2006*) knockout mice, indicating an absolute requirement for MuSK and LRP4. Our finding that APP-Fc, but not RAP-Fc, can interact with LRP4 and induce MuSK phosphorylation at low level independent of agrin is consistent with both observations. Interestingly, however, and in contrast to our results with LRP4, the intracellular domain of APP appears to be required for normal NMJ development (*Li et al., 2010*). This suggests a role of these highly conserved C-terminal sequences in either signaling with APP serving as a co-receptor in the MuSK signaling complex, or a role in regulating the trafficking of all or part of the components of this complex.

It is conceivable that promotion of AChR clustering by homo- or heteromeric interaction of APP with itself or APLP2, and through inclusion of LRP4 and MuSK, requires an additional protein that interacts with the APP cytoplasmic domain, and that this requirement resembles the need for the interaction of rapsyn with the AChR and MuSK (*Bromann et al., 2004*). In their study, Bromann and colleagues further showed that immobilized, but not soluble, agrin is able to aggregate MuSK and promote AChR clustering independent of MuSK's kinase activity, suggesting that MuSK scaffolding plays an important role in AChR clustering. Our findings show that APP is independently capable of promoting AChR clustering, and that it further enhances clustering in the presence of agrin. Moreover, agrin potentiates the interaction of APP with LRP4. Thus, it is conceivable that APP cooperates with agrin and LRP4 to establish such an extracellular scaffold to promote MuSK aggregation and AChR clustering. AChR clustering can also be mediated by a variety of agrin independent mechanisms, including treatment with neuraminidase and VVA lectin (*Martin and Sanes, 1995*; *Grow et al., 1999a,b*) and by defucosylation of muscle agrin, which unmask its ability to induce MuSK phosphorylation (*Kim et al., 2008a*).

In addition, we have also obtained further evidence showing functional interaction of LRP4, APP and agrin during NMJ formation. We have analyzed mutant mice that lack the *App* family member *Aplp2* on the background of homozygous LRP4 ECD (*Lrp4*^ECD/ECD^;*Aplp2*^–/–^). Like *App*, *Aplp2* is also required for normal NMJ formation, and mice lacking both *App* family members exhibit aberrant apposition of presynaptic marker proteins with postsynaptic acetylcholine receptors and excessive nerve terminal sprouting and a reduced number of synaptic vesicles at presynaptic terminals (*Wang et al., 2005*). This phenotype is consistent with the role of *App* family members in mediating trans-synaptic adhesion functions (*Wang et al., 2009*) through homo- and hetero-oligomeric interactions (*Soba et al., 2005*). Loss of both APP and APLP2, but not of either protein alone, might critically reduce this trans-synaptic interaction (*Figure 11*), resulting in aberrant apposition of nerve terminals and AChR clusters.

We did not observe such an aberrant apposition in our *Lrp4*^ECD/ECD^;*App*^–/–^ model or in *Lrp4*^ECD/ECD^;*Aplp2*^–/–^ mice. However, as in *Lrp4*^ECD/ECD^;*App*^–/–^, functional NMJ formation, as evidenced by greatly impaired postnatal survival (*Table 2*) and AChR clustering (*Figure 5*), is also significantly compromised in the *Lrp4*^ECD/ECD^;*Aplp2*^–/–^ double mutants. This would suggest that while APP and APLP2 are important for establishing trans-synaptic contact, LRP4 rather serves, through a hetero-oligomeric interaction with APP, MuSK and likely APLP2, to integrate agrin through cooperative binding into a spatially restricted signaling complex that focuses the molecular instructions required for AChR clustering at the surface of the muscle fiber.

**Table 2.** Survival analysis of *Lrp4*^ECD/ECD^;*Aplp2*^–/–^ double mutant pups

| Genotype | Expected | Observed | Chi-square test |
|---|---|---|---|
| *Lrp4*^+/ECD^;*Aplp2*^+/–^ | 68 | 82 | n.s. |
| *Lrp4*^+/ECD^;*Aplp2*^–/–^ | 69 | 75 | n.s. |
| *Lrp4*^ECD/ECD^;*Aplp2*^+/–^ | 58 | 51 | n.s. |
| *Lrp4*^ECD/ECD^;*Aplp2*^–/–^ | 54 | 17 | * |

n.s., not significant.
*Significant at p<0.01.
Pups were genotyped at weaning (~25 days of age) to determine postnatal survival rates.

This oligomeric interaction of the LRP4 ECD may explain the residual function of the nonmembrane-associated soluble receptor in the LRP4 ECD mutants. While membrane anchoring firmly restricts LRP4 to the two-dimensional surface of the muscle fiber and thereby ensures its continued presence in the MuSK/agrin/APP/APLP2 signaling complex, the loss of the transmembrane segment can be partially compensated for by the combined interactions of APP, APLP2, MuSK and agrin with the LRP4 ECD. The reduced signal strength that results in impaired NMJ size and number can be explained by the inevitable reduction of LRP4 ectodomain at the NMJ and relative dilution of secreted ECD throughout the muscle surface and the interstitial space (*Figure 10*). This in turn is partially compensated for by the increased expression of LRP4 ECD in the heterozygous and homozygous mutants (*Figure 10E,F*).

We have further found that the non-neural Z0 as well as the neural Z8 and Z19 splice forms can bind to APP (*Figure 8D*). Interaction of Z0 (muscle) agrin, which cannot efficiently promote AChR clustering at physiological levels (*Kim et al., 2008a*), may have nevertheless physiological importance in this context by negatively regulating the recruitment of the MuSK activating, neural agrin isoforms outside of nerve contact sites, where their concentrations would be reduced. In doing so, muscle agrin would serve to potently suppress the formation of aberrant signaling complexes by blocking LRP4 and APP family proteins with the inactive form. This inhibition would be overcome by the higher concentrations of neural agrin at the nerve terminals.

On another note, we have previously shown that LRP4 can interact with several modulators of the Wnt pathways, that is Wise, Dkk, and Sost (*Dietrich et al., 2010*; *Karner et al., 2010*). These modulators bind to the homologous β-propeller domain in LRP5. It is intriguing to speculate that these interactions can also contribute to the modulation of NMJ development or function.

In summary, the findings we have reported here reveal new insights into the molecular mechanisms that govern the formation of the neuromuscular synapse and that are consistent with the recent findings from the Burden and Mei laboratories (*Wu et al., 2012*; *Yumoto et al., 2012*; *Zong et al., 2012*). Furthermore, they raise interesting new questions about the interplay of agrin, MuSK, APP and LRP4 in the CNS where all these proteins are also expressed. There they might, in principle, work together in an analogous or similar manner to regulate synaptogenesis and promote neuronal survival. LRP4 belongs to an evolutionarily conserved class of ApoE receptors with essential functions in the regulation of neurotransmission at glutamatergic synapses (*Herz and Chen, 2006*). We have recently shown that ApoE in an AD-associated, isoform-specific manner impairs ApoE receptor recycling at central synapses (*Chen et al., 2010b*) and that this impairs the ability of ApoE receptor-dependent signals to prevent β-amyloid induced synaptic suppression (*Durakoglugil et al., 2009*). The findings we have reported here, together with those by Wang et al., who have proposed that perturbed APP synaptic adhesion activity may contribute to synaptic dysfunction and AD pathogenesis (*Wang et al., 2009*), reaffirm the NMJ as a versatile and useful experimental model system on which the cell biology of ApoE, APP and ApoE receptor trafficking and its pathobiology at peripheral and central synapses can be investigated.

## Materials and methods

Recombinant rat agrin C-terminal fragment, Ala1153 Pro1959 (Pro1788-Ser1798del), with an N-terminal Met and 6xHis tag (Cat. # 550-AG) was purchased from R&D Systems (Minneapolis, MN). $Lrp4^{ECD/ECD}$ and $Lrp4^{-/-}$ mice have been described previously (*Johnson et al., 2005*; *Karner et al., 2010*). Rabbit polyclonal anti-syntaxin and rabbit polyclonal anti-synaptotagmin 2 were gifts from Dr Thomas Südhof, Stanford University School of Medicine, Palo Alto, CA, and used at 1:1000 dilution of immunofluorescence. Rabbit polyclonal anti-neurofilament (NF150) is from Chemicon, Temecula, CA, and was used at 1:1500 dilution. Rabbit polyclonal anti-LRP4 extracellular domain antibody was generated in the Kröger lab against the third and fourth beta-propeller domain of mouse LPR4 and was used at 1:100 dilution for immunofluorescence. Rabbit polyclonal anti-LRP4 antibody directed against the carboxyl-terminus of LRP4 was generated in the Herz lab and used for immunoblotting at 1:1000 dilution. Rabbit polyclonal anti-APP was generated in the Herz lab against the carboxyl-terminus of LRP4 and was used at 1:500 dilution for immunofluorescence.

### Generation of mutant mice and animal husbandry

The generation of $App^{-/-}$ (*Zheng et al., 1995*) and $Lrp4^{ECD/ECD}$ (*Johnson et al., 2005*) mice has been reported. $Lrp4^{\Delta ICD/\Delta ICD}$ mice were generated using the same targeting construct, except that sequences containing the LRP4 transmembrane segment followed by a Myc-epitope were introduced in place of

the stop codon (*Johnson et al., 2006*). The Lrp4[ΔICD/ΔICD] knockin mice were generated using a replacement vector based on the construct described for the LRP4 ECD mutant. The same long arm of homology, but lacking the introduced premature stop codon, was used. A cDNA insert expressing the transmembrane segment followed by a Myc epitope was cloned using an upstream Bst1107I site in the long arm and a BsrGI site in the bovine growth hormone 3'UTR. The oligonucleotide KI5 (5'-GTATACTG CTGATTTTGTTGGTGATCGCGGCTTTG-3') was used as the 5' primer and MEJ380 (5'-GTGTGTTGTAC ATCAGCTATTCAGATCCTCTTCTGAGATGAGTTTTTGTTCCTTGGATTTCCTGTGTCTGTATAGCATC AAAG-3') was used as the 3' primer to amplify the cDNA insert for the TM-Myc Epitope cassette.

*Lrp4* null mice were generated as described in *Figure 1* by replacing the first exon with a neomycin resistance cassette. The long arm of homology upstream of the first exon of *Lrp4* was generated by PCR amplification using the primers MEJ23 (5'-GCGGCCGCCAGGTCATGAAGTGAGTGCTGAGCCA CTGGG-3') and MEJ24 (5'-CCACCACCGCCTCATGGTGCTGCGGCCGCC-3'). The short arm of homology downstream of the first exon of *Lrp4* was generated by PCR amplification using the primers MEJ33 (5'-CTCGAGGAGCGGTCTGCAGATCCTGGCGATTCACGG-3') and MEJ35 (5'-CTCGAGGGT TACAGACTCTGCAACTGCTCTACCTCATTG-3'). The long arm and short arm of homology were cloned into pJB1 using the NotI and XhoI restriction sites, respectively.

Animals were maintained on a mixed 129SvEv Bradley;C57BL/6J background by heterozygous intercrossing. Wild-type (WT) control mice were obtained from the same crosses. Animals were maintained on 12-hr light/12-hr dark cycles and fed a standard rodent chow diet (Diet 7001; Harlan Teklad, Madison, WI) and water ad libitum. No sexual dimorphism of phenotype was observed. All procedures were performed in accordance with the protocols approved by the Institutional Animal Care and Use Committee of the University of Texas Southwestern Medical Center at Dallas.

## Production and purification of recombinant proteins

A cDNA encoding the LRP4 ligand binding domain (LBD, aa 27–349, accession number CAM24075) was inserted downstream of the maltose-binding protein (MBP) coding sequence into pMAL-p4x (NEB). pMAL-p4x and pMAL-LRP4LBD-p4x constructs were expressed in *Escherichia coli* (BL21) to produce properly folded MBP and MBP-LRP4LBD fusion protein in the periplasm. Periplasmic extracts were subjected to amylose column (NEB, Ipswich, MA) to obtain affinity-purified MBP and MBP-LRP4LBD.

Fc fusion proteins were generated by fusing the extracellular domains of mouse LRP4 (residues 1–1650, LRP4-Fc), APP695 (residues 1–596, APP-Fc), or full length RAP (RAP-Fc) to the constant region of human IgG (Fc). Secreted fusion proteins were produced by transfecting HEK293A cells with pCDNA3.1-LRP4-Fc, pCDNA3.1-APP-Fc, or pCDNA3.1-RAP-Fc constructs using FuGENE 6 (Roche, Indianapolis, IN). Fc fusion proteins secreted into media were collected and purified on Protein A-Sepharose columns (Sigma, St. Louis, MO). APP-FLAG was generated by fusing 3xFLAG in place of Fc to the carboxyl-terminus of APP-695 (residues 1–596) in pCDNA3.1.

## Protein interaction studies

To investigate binding between LRP4 and APP, 100 ng of purified APP-Fc was incubated with Protein A-agarose beads (Sigma) in phosphate-buffered saline (PBS) containing 0.1% bovine serum albumin (BSA) for 4 hr at 4°C to generate Fc-agarose conjugates. The conjugates were then incubated with 100 ng of purified MBP or MBP-LRP4 LBD in PBS containing 0.1% BSA for 4 hr at 4°C. Protein coprecipitated with APP-Fc was detected by Western blotting with anti-MBP antibody (NEB), and the membrane was reprobed with anti-human IgG (Fc specific) antibody (Sigma) to determine the levels of APP-Fc.

To investigate binding between agrin and LRP4 or APP, 1.5 ml of LRP4-Fc, APP-Fc, or RAP-Fc conditioned medium was incubated with Protein A-agarose beads for 4 hr at 4°C to generate Fc-agarose conjugates. The conjugates were then incubated with 1 µg of neuronal agrin (R&D Systems) in Dulbecco's modified Eagle's medium (DMEM) containing 0.1% BSA for 4 hr at 4°C. Agrin coprecipitated with Fc fusion protein was detected by Western blotting with anti-agrin antibody (R&D Systems). The amount of Fc fusion protein present in the reaction was determined by reprobing the membrane with an anti-Fc antibody.

To investigate the role of agrin in LRP4–APP interaction, 100 ng of purified APP-Fc were conjugated to Protein A-agarose beads in PBS containing 0.1% BSA for 4 hr at 4°C. The conjugates were then incubated with 100 ng of purified MBP-LRP4LBD and 0–1000 ng of neuronal agrin in PBS containing 0.1% BSA for 4 hr at 4°C. The precipitates were then subjected to anti-MBP, anti-Agrin, or anti-Fc immunoblotting.

To investigate the differential interactions of Agrin isoforms with APP, we used constructs encoding the C-terminal 110 kDa fragments of agrin isoforms Z0, Z8 and Z19 containing N-terminal Flag tag and C-terminal Myc and 6xHis tags, respectively (*Burgess et al., 2000*; *Bogdanik and Burgess, 2011*). These plasmids were generously provided by Dr Robert Burgess (Jackson Laboratory, Bar Harbor, ME). Agrin fragments were produced as secreted proteins in HEK293A cells grown in DMEM containing 0.1% BSA. 100 ng of purified APP-Fc or RAP-Fc was incubated with 300 μl of agrin containing supernatants overnight at 4°C followed by adsorption to Protein A Dynabeads (Invitrogen, Grand Island, NY) for 10 min at room temperature. After magnetic isolation, bound proteins and supernatants were analyzed by Western blotting using an anti-Myc antibody (9E10). Equal binding of the respective Fc proteins to the Dynabeads was detected by Western blotting using an anti-human IgG (Fc specific) antibody (Sigma, St. Louis, MO).

## Co-immunoprecipitation of LRP4–APP–MuSK complex from muscle

Hind limb muscle proteins in WT embryo (E14.5) were extracted in lysis buffer (PBS supplemented with 5 mM EDTA, 5 mM EGTA, 1% digitonin, and completed protease inhibitor tablet [Roche]). After centrifugation at 18,000×$g$ for 5 min, the supernatant was incubated with 10 μl of an affinity purified anti-LRP4, anti-ApoER2, or anti-LRP1 antibody overnight at 4°C. Each antibody and bound proteins were absorbed to Protein A Dynabeads (Invitrogen) for 10 min at room temperature. After magnetic isolation, bound proteins and supernatants were analyzed by immunoblotting.

## Cell aggregation assay

HEK293T cells were co-transfected for 48 hr with a 3:1 ratio of pDsRed:pCDNA3.1-LRP4 full-length constructs or of pEGFP:pCDNA3.1-APP full-length constructs to generate LRP4-expressing red-fluorescent and APP-expressing green-fluorescent cells, respectively. HEK293T cells transfected with pDsRed or pEGFP vector were used as control cells: red- or green-fluorescent cells. Cells were incubated with 0.05% Trypsin/0.53 mM EDTA in calcium- and magnesium-free Hank's balanced salt solution (HBSS) for 5 min followed by trituration, washed twice with HBSS and resuspended in aggregation buffer (0.1g glucose, 0.1 g BSA, 0.26 g HEPES, 13.75 mg $CaCl_2$, and 10 mg DNase I in 100 ml HBSS). LRP4 and APP expression was determined from single cell suspensions by immunoblotting. LRP4-expressing red-fluorescent and APP-expressing green-fluorescent cells or red- and green-fluorescent cells were mixed and mutated at 37°C for 30 min in 16-mm wells precoated with 1% BSA. Aggregation was stopped by fixation with 2.5% glutaraldehyde. Cell suspensions were drop-plated and covered with coverslips. Fluorescent cell images were captured with a Zeiss 10x/0.30 NA dry objective on a Zeiss Axioplan 2 microscope. Aggregates of >3000 μm$^2$ in random fields were scored.

## Immunofluorescence analysis

Morphological analysis of the NMJ was carried out in both diaphragm and triangularis sterni muscles using procedures described previously (*Liu et al., 2008*). Briefly, diaphragm (E14.5, E16.5, E18.5 or P12) or triangularis sterni (E18.5, P12 or 3-month-old) muscles were dissected out, fixed with 2% paraformaldehyde in 0.1 M phosphate buffer (pH 7.3) overnight at 4°C, washed thoroughly with PBS, and incubated with 0.1 M glycine in PBS for 30 min. Samples were incubated with Texas Red-conjugated α-bungarotoxin (2 nM, Molecular Probes) in antibody dilution buffer (0.01 M phosphate buffer, 500 mM NaCl, 3% BSA and 0.01% thimerosal) for 30 min at room temperature to label postsynaptic AChR, washed with PBS and incubated with rabbit polyclonal anti-neurofilament (1:1500; Chemicon, Billerica, MA) and rabbit polyclonal anti-synaptotagmin 2 (1:1000, a generous gift from Dr Thomas Südhof, Stanford University School of Medicine, Palo Alto, CA) antibodies diluted in antibody dilution buffer overnight at 4°C to stain presynaptic motor axons. For sections, samples were incubated in antibodies against LRP4 (rabbit polyclonal, 1:100) or APP (rabbit polyclonal, 1:500) overnight at room temperature. After washing three times with 0.5% Triton X-100 in PBS, samples were incubated with FITC-conjugated anti-rabbit IgG overnight at 4°C, washed in PBS, and mounted in Vectashield mounting medium. Fluorescent images were captured using a Zeiss LSM 510 confocal microscope. Quantitative measurements of AChR cluster number and size were made using NIH ImageJ software. For the analysis of AChR cluster number, Texas Red-labeled AChR clusters on the same right ventral region of each diaphragm were counted. For the analysis of AChR size, images acquired at high magnification were used.

To determine LRP4 distribution, hindlimb muscles from E14.5, E16.5 and E18.5 embryos were transversely sectioned (20 μm). Sections were incubated with 2 nM of Texas Red-conjugated α-bungarotoxin for 30 min at room temperature to label postsynaptic AChR. Then, sections were incubated with rabbit

polyclonal antibody against LRP4 extracellular domain (1:100) overnight at 4°C. After washing with 0.1% Triton X-100 in PBS, the samples were incubated with fluorescein isothiocyanate-conjugated anti-rabbit IgG for 2 hr at room temperature, washed with PBS and mounted in Vectashield mounting medium. Images were acquired under an upright fluorescence microscope (Olympus BX51) using a Hamamatsu ORCA-285 camera, and fluorescence intensity (mean gray value) was measured using NIH ImageJ.

## 3D reconstruction of confocal images

Images were captured using a Zeiss LSM 510 META confocal microscope with a Zeiss 40x/1.3 NA oil immersion objective. Excitation wavelengths used for the red and green channels were 543 nm (HeNe1 Laser) and 488 nm (Argon Laser), respectively. Z-stacks were acquired at an optical slice interval of 0.6 µm. The Reuse feature from the Zeiss LSM acquisition software allowed capturing of all images at the same laser power, detector gain and amplifier offset for all specimens. 3D reconstruction of the Z-stack images was done using Imaris software (Bitplane Inc, MN). Initial image analysis included applying the background subtraction algorithm in Imaris. The intensity for the green and red channels was adjusted in Blend mode during volume rendering and an Iso Surface model was created for the background-subtracted images.

## Electron microscopy

Ultrastructural analysis of the NMJ was carried out as previously described (*Liu et al., 2009*; *Chen et al., 2010a*). Deeply anesthetized animals were fixed via cardiac perfusion with a mixture of 1% glutaraldehyde and 4% paraformaldehyde in 0.1 M phosphate buffer, pH 7.4. Lumbrical muscles were dissected and post-fixed in the same solution overnight at 4°C. Tissues were then rinsed with the 0.1 M phosphate buffer, trimmed to small pieces and post-fixed with 1% osmium tetroxide for 3 hr on ice. Tissues were then dehydrated in a graded series of ethanol, infiltrated and polymerized in Epon 812 (Polysciences, Warrington, PA). Epon blocks were cut at 1 µm and stained with toluidine blue for light microscopic observation. Ultrathin (70 nm) sections were mounted on Formvar-coated grids and stained with uranyl acetate and lead citrate. Electron micrographs were acquired using a Tecnai (Netherlands) electron microscope operated at 120 kV.

## AChR clustering assay

Mouse C2C12 cells maintained as undifferentiated myoblasts in DMEM supplemented with 20% fetal bovine serum and 0.5% chick embryo extract were seeded on 8-well plastic chamber slides (Nalge Nunc International, Rochester, NY) and grown to ~90% confluence. Myoblasts were fused into myotubes in differentiation medium (DMEM supplemented with 2% horse serum) for 3 days. Myotubes were incubated with 300 µl differentiation medium containing 3 µl Protein A-agarose beads (50% slurry in PBS containing 0.1% BSA) conjugated without or with saturating amounts of APP-Fc or RAP-Fc for 24 hr in the absence or presence of 0.1 ng/ml agrin. At the end of the incubation, myotubes were fixed with 1% paraformaldehyde in 0.1 M phosphate buffer for 30 min, washed with PBS and incubated with 2 nM of Texas-Red conjugated α-bungarotoxin (Molecular Probes, Grand Island, NY) for 30 min to label AChRs. Myotubes were then washed with PBS and mounted in Vectashield mounting medium. For each sample, 50 fluorescent images were randomly captured using a Hamamatsu ORCA-285 camera under 400× magnification. A cluster with a length >2 µm was defined as a patch of AChRs. All the myotubes that had AChR clusters in the fields were analyzed blindly, and the size and number of AChR clusters/200 µm myotube length were measured using NIH ImageJ software and normalized to control levels.

## Primary myotube culture

Muscles were quickly removed from the limbs of WT, *Lrp4*−/−, and *Musk*−/− embryos (E18.5) in Hanks' Balanced Salt Solution (Invitrogen). Muscles were minced into small pieces before enzymatic digestion for 45 min at 37°C in HBSS supplemented with 2 mg/ml of Type II collagenase (Worthington) and 0.4 mg/ml of DNAse I (Worthington, Lakewood, NJ). Dissociated muscle cells were collected by centrifugation at 2000×*g* for 5 min, resuspended in culture medium (DMEM supplemented with 10% horse serum, 5% fetal calf serum and 1% chick embryo extracts), and filtered through a Cell Strainer with diameter of 100 µm (BD Falcon, San Jose, CA). Cells were then seeded on a 100-mm culture dish and incubated in a 37°C, 5% $CO_2$ incubator for 2 hr in order to attach and remove fibroblasts. Suspended myoblasts were collected by centrifugation at 2000×*g* for 5 min and seeded on 8-well plastic chamber slides (Fisher, Pittsburgh, PA). After growing cells in the culture medium until approximately 90% confluence,

the culture medium was replaced with differentiation medium (DMEM supplemented with 2% horse serum), and this medium was changed every day for 3 days to fuse myoblasts into myotubes for the AChR clustering assay as described.

## MuSK phosphorylation assay

Differentiated myotubes were incubated with indicated concentrations of agrin, APP-Fc, or RAP-Fc for 30 min at 37°C. The myotubes were rinsed once on ice with PBS containing 1 mM sodium orthovanadate and 50 mM sodium fluoride and extracted in lysis buffer (30 mM triethanolamine, 1% NP-40, 50 mM NaCl, 5 mM EDTA, 5 mM EGTA, 50 mM sodium fluoride, 2 mM sodium orthovanadate, 1 mM sodium tetrathionate, 1 mM N-ethylmaleimide, 10 µM Pepstatin, 0.5 mg/ml Pefabloc, and complete protease inhibitor tablet [Roche]). Lysates were centrifuged at 18,000×$g$ for 5 min and the supernatant was used as total myotube extracts. To immunoprecipitate MuSK, the extracts were incubated overnight at 4°C with anti-MuSK antibody (Abcam, Cambridge, MA) followed by precipitation of the antibody with Protein A/G PLUS-Agarose (Santa Cruz, Dallas, TX). Precipitated MuSK was separated by SDS-PAGE and the phosphorylation (activation) status of MuSK was determined by Western blotting using anti-phosphotyrosine antibody 4 G10 Platinum (Millipore, Billerica, MA). Molarities are stated for monomers.

## Statistical analyses

Data are presented as mean ± standard error of the mean (SEM). Statistical analyses of differences between or among control, mutant and treatment groups were carried out using Student's $t$ test or one-way analysis of variance (ANOVA). Post hoc test was carried out by using Tukey analyses when a significant F-value was obtained in ANOVA. A p value of $<0.05$ was considered to be significant.

## Acknowledgements

We are indebted to Gary Philips for critical reading of the manuscript, and to Mike Brown and Joe Goldstein for helpful suggestions. We thank Robert Burgess for agrin constructs, Wen-Ling Niu, Huichuan Reyna, Priscilla Rodriguez, Rebekah Hewitt, Vicki Kerr, and Isaac Rocha for excellent technical assistance, and Nancy Heard for invaluable help with artwork.

## Additional information

### Funding

| Funder | Grant reference number | Author |
| --- | --- | --- |
| National Institutes of Health | HL20948, HL63762, NS055028 | Weichun Lin, Joachim Herz |
| American Health Assistance Foundation | | Joachim Herz |
| Lupe Murchison Foundation | | Joachim Herz |
| Consortium for Frontotemporal Dementia Research | | Joachim Herz |
| Deutsche Forschungsgemeinschaft | SFB780 | Joachim Herz |
| Alexander von Humboldt Stiftung | | Joachim Herz |
| Brightfocus Foundation | | Joachim Herz |
| Alzheimer's Disease Center | | Yun Liu |
| Cain Foundation in Medical Research | | Weichun Lin |

The funders had no role in study design, data collection and interpretation, or the decision to submit the work for publication.

### Author contributions

HYC, YL, Conception and design, Acquisition of data, Analysis and interpretation of data; CT, EBJ, Acquisition of data; YS, Acquisition of data, Analysis and interpretation of data; AK, SK, Generated the unpublished antibody against the LRP4 extracellular domain; REH, Performed all transgene manipulations in vivo; WL, JH, Conception and design, Analysis and interpretation of data, Drafting or revising the article

## Ethics

Animal experimentation: The animal experiments described in this paper were reviewed and approved by the Institutional Animal Care and Use Committee (IACUC) at UT Southwestern where the experiments were exclusively conducted. The animal assurance number is A3472-01. All experiments were conducted along the guidelines described in "The Guide for the Care and Use of Laboratory Animals": Eighth Edition; Committee for the Update of the Guide for the Care and Use of Laboratory Animals; National Research Council; ISBN: 0-309-15401-4, 248 pages, 6 x 9, (2010). This PDF is available from the National Academies Press at: http://www.nap.edu/catalog/12910.html.

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
