## [Decision Letter]

Thank you for choosing to send your work entitled “APP Interacts with LRP4 and Agrin to Coordinate the Development of the Neuromuscular Junction in Mice” for consideration at *eLife*. Your article has been evaluated by a Senior editor, a Reviewing editor, and 3 reviewers. The Reviewing editor has assembled the following comments based on the reviewers' reports.

This is an interesting manuscript that reports the novel finding that APP participates in the signaling that underlies the formation of the neuromuscular junction (NMJ), together with agrin, LRP4 and MuSK. Although the role of APP in this context was suggested in several previous publications, this is the first study that combines molecular genetics, cell biology and biochemical methods to delineate its involvement in agrin-LRP4-MuSK signaling. A number of points need to be addressed by the authors to substantiate the model they propose.

1) The authors propose that APP participates in the anchoring of the LRP4 extra-cellular domain at the NMJ. However, although the APP/LRP4 interaction is shown and although the deletion of APP worsens the LRP4 ECD phenotype, no direct evidence supports this hypothesis. The demonstration would be more convincing if muscle immunostaining showed the localization of LRP4 ECD at the NMJ in the presence and absence of APP. One would expect LRP4 ECD accumulation to decrease in the absence of APP. Alternatively LRP4ECD-GFP fusions could be expressed in WT and APP^-/-^ muscles. LRP4 ECD retention at the membrane, and/or colocalization with MuSK, could also be evaluated in cultured myotubes expressing APP, or not.

It would also be informative to know more about the expression patterns of APP and LRP4. A previous study suggested that the expression of APP in muscles starts at E16 and is concentrated at NMJs progressively ([1], MCN 15:355). It is unclear if APP and LRP4 are expressed in a spatially and temporally correlated manner. The authors should provide a developmental expression profile of these two proteins in both motoneurons and muscles.

2) In vitro studies seem to suggest that APP is sufficient to induce AChR clusters in cultured muscles. However, there was no discernible phenotype of the APP^-/-^ mutant, until the transmembrane domain of LRP4 was truncated. Because APP could bind both muscle and neuronal agrin, an interesting question is if APP acts by aggregating agrin. The conclusion in the text and in the model (Figure 10) that APP-LRP4 interaction in the absence of agrin can activate MuSK to promote AChR clustering is not supported by the data shown. C2C12 cells express muscle agrin, which binds to APP as shown in Figure 9. APP's effect could be studied in agrin^-/-^ or KD muscle fibers. Alternatively, if APP overexpression could rescue the agrin null phenotypes in vivo that would be ever better. The authors should also check ectopic AChR cluster formation induced by agrin under LRP4^ECD/ECD^ and LRP4^ECD/ECD^ APP^-/-^ conditions; again, this could be done in isolated muscle fibers.

3) The authors propose that APP could constitute the factor through which neuromuscular contacts initially form independently of agrin. This hypothesis is very interesting but requires additional proof, such as showing that, in the absence of agrin, APP is required for these initial contacts to form. It would be time consuming to cross APP^-/-^ and agrin^-/-^ mice, but the authors could already evaluate the implication of APP and APLP2 in the initial steps of neuromuscular junction formation by evaluating the requirement of these proteins for the prepatterning of AChR clusters that precedes nerve-muscle contact, i.e., by analyzing muscle prepatterning in APP^-/-^ and APLP2^-/-^ mice.

4) The demonstration of the interaction between APP and LRP4 is strong and convincing. In light of the recent results published by Steve Burden and colleagues showing that LRP4 constitutes the attraction signal for motoneurones, it is tempting to speculate from the present data that APP could constitute the neuronal ligand of LRP4. Primary motoneuron cultures from APP^-/-^ mice could be cocultured with LRP4 beads to challenge this hypothesis.

5) In one set of binding assays, the LDB fragment was used. There was no data to show that this domain was required for fl LRP4 interaction with APP. This could be tested in experiments described in Figure 7. Furthermore there were no data that showed the LRP4-APP interaction was necessary for NMJ formation or maturation. Ideally it should be shown in vivo that LRP4 without the LBD cannot rescue, but LRP4 wt can rescue the null phenotypes. At the very least, in vitro studies should be performed to see if APP-induced AChR clustering and MuSK activation requires the LBD. Moreover, description of this domain should be included in the Results rather than within the Methods. This domain, aa 27–349, has been shown to interact with MuSK. Perhaps the authors can comment in the Discussion on whether this domain interacts with WISE, DKK, and SOST?

*Other points of presentation that need attention*:

a) The conclusion that “membrane anchoring of the receptor is required to maintain the level of signal strength necessary for postnatal maturation of the synapse” seems to be a misinterpretation. Many “maturation” events at ECD/ECD NMJs were apparently normal, including synapse elimination, subunit switch, etc.

b) There is some concern regarding the conclusion that “the failure of RAP-Fc to induce AChR clustering suggests that LRP4 dimerization alone is not sufficient, but that possibly the formation of an extracellular scaffold”. No data to show that RAP-Fc binds to LRP4 is presented in this paper. It binds to ApoE, but this does not suggest that it also binds LRP4. If it does, where in LRP4 does it bind? The LBD?

c) Based on the fluorescent images presented in Figure 6, the impairment of postsynaptic differentiation, as illustrated by AChR clusters, in APP/LRP4^ECD^ double mutant is subtle. A quantitative analysis is required to show if the double mutant significantly affects postsynaptic differentiation.

---

## [Author Response]

*The authors propose that APP participates in the anchoring of the LRP4 extra-cellular domain at the NMJ. However, although the APP/LRP4 interaction is shown and although the deletion of APP worsens the LRP4 ECD phenotype, no direct evidence supports this hypothesis. The demonstration would be more convincing if muscle immunostaining showed the localization of LRP4 ECD at the NMJ in the presence and absence of APP. One would expect LRP4 ECD accumulation to decrease in the absence of APP. Alternatively LRP4ECD-GFP fusions could be expressed in WT and APP*^*-/-*^
*muscles. LRP4 ECD retention at the membrane, and/or colocalization with MuSK, could also be evaluated in cultured myotubes expressing APP, or not*.

*It would also be informative to know more about the expression patterns of APP and LRP4. A previous study suggested that the expression of APP in muscles starts at E16 and is concentrated at NMJs progressively (*[1]*, MCN 15:355). It is unclear if APP and LRP4 are expressed in a spatially and temporally correlated manner. The authors should provide a developmental expression profile of these two proteins in both motoneurons and muscles*.

We thank the reviewers for this excellent suggestion. We have carried out immunostaining using antibodies against the LRP4 ectodomain. As expected and envisioned by the reviewer, we found that anti-LRP4 immunostaining was more diffuse in LRP4^ECD/ECD^ and in APP/APLP2 double knockout muscle compared to WT muscle (Figure 10). These results impressively show that LRP4 and APP family members are co-expressed and stabilize each other at the NMJ, as also demonstrated by Figure 8.

We have investigated the localization of LRP4 and APP during NMJ synaptogenesis at E14.5, E16.5 and E18.5, using anti-LRP4 and anti-APP antibodies. We found that both LRP4 and APP were localized at the NMJ at E18.5. However, at E14.5 or E16.5 the staining was diffusely distributed across the muscle surface. These requested data are now included in the manuscript as part of Figure 10. Additionally, we have carried out immunostaining on embryonic spinal cord sections (E14.5, E16.5 and E18.5), using anti-LRP4 or anti APP antibodies. However, we were unable to detect specific staining for both antibodies in the spinal cord at all three stages.

*2) In vitro studies seem to suggest that APP is sufficient to induce AChR clusters in cultured muscles. However, there was no discernible phenotype of the APP*^*-/-*^
*mutant, until the transmembrane domain of LRP4 was truncated. Because APP could bind both muscle and neuronal agrin, an interesting question is if APP acts by aggregating agrin. The conclusion in the text and in the model (*Figure 10*) that APP-LRP4 interaction in the absence of agrin can activate MuSK to promote AChR clustering is not supported by the data shown. C2C12 cells express muscle agrin, which binds to APP as shown in*
Figure 9*. APP's effect could be studied in agrin*^*-/-*^
*or KD muscle fibers. Alternatively, if APP overexpression could rescue the agrin null phenotypes* in vivo *that would be ever better. The authors should also check ectopic AChR cluster formation induced by agrin under LRP4*^*ECD/ECD*^
*and LRP4*^*ECD/ECD*^
*APP*^*-/-*^
*conditions; again, this could be done in isolated muscle fibers*.

We agree with the reviewer that these would be informative experiments. For the time being, they have not given us conclusive in vitro results. We will need to pursue this point further by generating agrin^-/-^;APP^-/-^;APLP2^-/-^ triple mutant mice. This is a challenge magnified by the low viability of the compound mutants. We have toned down our statements in the text and the discussion accordingly.

*3) The authors propose that APP could constitute the factor through which neuromuscular contacts initially form independently of agrin. This hypothesis is very interesting but requires additional proof, such as showing that, in the absence of agrin, APP is required for these initial contacts to form. It would be time consuming to cross APP*^*-/-*^
*and agrin*^*-/-*^
*mice, but the authors could already evaluate the implication of APP and APLP2 in the initial steps of neuromuscular junction formation by evaluating the requirement of these proteins for the prepatterning of AChR clusters that precedes nerve-muscle contact, i.e., by analyzing muscle prepatterning in APP*^*-/-*^
*and APLP2*^*-/-*^
*mice*.

We thank the reviewers for this excellent point. We have initiated the experiment by breeding agrin mutant mice with APP and APLP2 mutant mice. Because of the redundancy of APP and APLP2, we will need to study agrin/APP/APLP2 triple mutant mice to properly evaluate the effect. Due to the time required for this extensive breeding we have not obtained the triple mutant mice. However, we have evaluated the implication of APP and APLP2 in the prepatterning of AChR clusters in E14.5 APP/APLP2 double mutant muscle. As shown in Figure 9, AChRs are clustered along the central region of E14.5 APP/APLP2 double mutant muscle. Intriguingly, individual AChR clusters appear less robust compared with those in the wild type, APP null or APLP2 null muscles. These results demonstrate that prepatterned AChR clustering can occur in the absence of APP and APLP2.

*4) The demonstration of the interaction between APP and LRP4 is strong and convincing. In light of the recent results published by Steve Burden and colleagues showing that LRP4 constitutes the attraction signal for motoneurones, it is tempting to speculate from the present data that APP could constitute the neuronal ligand of LRP4. Primary motoneuron cultures from APP*^*-/-*^
*mice could be cocultured with LRP4 beads to challenge this hypothesis*.

We thank reviewers for this insightful comment. It is interesting that the results published by Steven Burdenʼs group (Yumoto, et al., Nature 2012) and Mei Linʼs group (Wu, et al., Neuron 2012) have shown that LRP4 expressed in non-muscle cells can induce presynaptic differentiation by aggregating presynaptic markers such as synapsin and SV2. Furthermore, studies from Lin Meiʼs group (Wu et al., Neuron, 2012) have shown that muscle specific deletion of LRP4 impairs both pre and postsynaptic development of the NMJ and that LRP4 in motor neurons is necessary for presynaptic differentiation of motor axons. It is tempting to speculate that APP, APLP2, or both, may serve as the neuronal ligand for LRP4 in inducing presynaptic differentiation. However, in vivo evidence indicates that APP/APLP2 is unlikely to be the major player in inducing presynaptic differentiation, because presynaptic differentiation is only mildly affected in neuronal specific deletion of APP/APLP2 (Wang et al., J Neuroscience 2009).

*5) In one set of binding assays, the LDB fragment was used. There was no data to show that this domain was required for fl LRP4 interaction with APP. This could be tested in experiments described in*
Figure 7*. Furthermore there were no data showed that the LRP4-APP interaction was necessary for NMJ formation or maturation. Ideally it should be shown* in vivo *that LRP4 without the LBD cannot rescue, but LRP4 wt can rescue the null phenotypes. At the very least,* in vitro *studies should be performed to see if APP- induced AChR clustering and MuSK activation requires the LBD. Moreover, description of this domain should be included in the Results rather than within the Methods. This domain, aa 27-349, has been shown to interact with MuSK. Perhaps the authors can comment in the Discussion on whether this domain interacts with WISE, DKK, and SOST*?

We show that the LBD fragment is sufficient for interaction with APP. We therefore did not test specifically whether the β-propeller portion of LRP4 also contributes to the interaction or not. Although this would ultimately provide additional structural detail to the demonstrated interaction between LRP4 and APP, it would not change the biological significance of the present study or the conclusions. We agree that in a future study one might address whether the interaction of LRP4, APP, agrin and MuSK can also be regulated by the various signal modulators (Wise, Dkk, Sost) that we have previously shown to bind to LRP4. Incidentally, Wise, Dkk and Sost all interact with the homologous β-propeller domain in LRP5/6. We ask for the reviewers’ and the editors’ understanding that to conclusively address this question would be more suitable for a full-fledged study in its own right.

Other points of presentation that need attention:

*a) The conclusion that “membrane anchoring of the receptor is required to maintain the level of signal strength necessary for postnatal maturation of the synapse” seems to be a misinterpretation. Many “maturation” events at ECD/ECD NMJs were apparently normal, including synapse elimination, subunit switch, etc*.

We thank the reviewers for these clarifying comments. Synapse elimination and subunit switch proceed normally at the NMJ in the LRP4ECD/ECD mice and we have revised the manuscript accordingly.

*b) There is some concern regarding the conclusion that “the failure of RAP- Fc to induce AChR clustering suggests that LRP4 dimerization alone is not sufficient, but that possibly the formation of an extracellular scaffold”. No data to show that RAP- Fc binds to LRP4 is presented in this paper. It binds to ApoE, but this does not suggest that it also binds LRP4. If it does, where in LRP4 does it bind? The LBD*?

We apologize for omitting the appropriate citation. This is now included (46). Note: RAP does not bind to ApoE. It universally binds to the ligand binding domains of ApoE receptors, i.e. the LDL receptor gene family. The structural basis for this has been nicely worked out by Steve Blacklow and colleagues (Structure of an LDLR-RAP Complex Reveals a General Mode for Ligand Recognition by Lipoprotein Receptors. Fisher C, Beglova N, Blacklow SC. Mol Cell. 2006 Apr 21;22(2):277-83). This reference has now also been cited.

*c) Based on the fluorescent images presented in*
Figure 6*, the impairment of postsynaptic differentiation, as illustrated by AChR clusters, in APP/LRP4 ECD double mutant is subtle. A quantitative analysis is required to show if the double mutant significantly affects postsynaptic differentiation*.

These data have been quantified and statistically analyzed. They are now shown in Figure 4 panel C.